# Self-propelled assembly of nanoparticles with self-catalytic regulation for tumour-specific imaging and therapy

Mengmeng Xia[1,8], Qiyue Wang[2,8], Yamin Liu [ID][2,8], Chunyan Fang[2,8], Bo Zhang [ID][2,3], Shengfei Yang[4], Fu Zhou [ID][1], Peihua Lin [ID][2], Mingzheng Gu[1], Canyu Huang[2], Xiaojun Zhang [ID][1], Fangyuan Li[4,5,6] ✉, Hongying Liu [ID][7] ✉, Guangfeng Wang[1] ✉ & Daishun Ling [ID][2,3] ✉

Targeted assembly of nanoparticles in biological systems holds great promise for disease-specific imaging and therapy. However, the current manipulation of nanoparticle dynamics is primarily limited to organic pericyclic reactions, which necessitate the introduction of synthetic functional groups as bioorthogonal handles on the nanoparticles, leading to complex and laborious design processes. Here, we report the synthesis of tyrosine (Tyr)-modified peptides-capped iodine (I) doped CuS nanoparticles (CuS-I@P1 NPs) as self-catalytic building blocks that undergo self-propelled assembly inside tumour cells via Tyr-Tyr condensation reactions catalyzed by the nanoparticles themselves. Upon cellular internalization, the CuS-I@P1 NPs undergo furin-guided condensation reactions, leading to the formation of CuS-I nanoparticle assemblies through dityrosine bond. The tumour-specific furin-instructed intracellular assembly of CuS-I NPs exhibits activatable dual-modal imaging capability and enhanced photothermal effect, enabling highly efficient imaging and therapy of tumours. The robust nanoparticle self-catalysis-regulated in situ assembly, facilitated by natural handles, offers the advantages of convenient fabrication, high reaction specificity, and biocompatibility, representing a generalizable strategy for target-specific activatable biomedical imaging and therapy.

Realizing in vivo manipulation of the dynamic performance of nanoparticles is a coveted goal at the intersection of nanotechnology and biomedical research, with significant implications for precise and intelligent nanotechnology-enabled diagnostics and biomedicine[1–5].

Among the diverse strategies pursued, the introduction of exogenous reactions for the construction of covalent bonds has emerged as an intriguing approach. This strategy capitalizes on specially designed and meticulously incorporated functional groups, such as alkynyl,

[1]School of Chemistry and Materials Science, Anhui Province Key Laboratory of Biomedical Materials and Chemical Measurement, Center for Nano Science and Technology, Anhui Normal University, 241000 Wuhu, China. [2]Frontiers Science Center for Transformative Molecules, School of Chemistry and Chemical Engineering, School of Biomedical Engineering, National Center for Translational Medicine, State Key Laboratory of Oncogenes and Related Genes, Shanghai Jiao Tong University, 200240 Shanghai, China. [3]World Laureates Association (WLA) Laboratories, 201203 Shanghai, China. [4]Institute of Pharmaceutics, Hangzhou Institute of Innovative Medicine, College of Pharmaceutical Sciences, Zhejiang University, 310058 Hangzhou, China. [5]Key Laboratory of Precision Diagnosis and Treatment for Hepatobiliary and Pancreatic Tumor of Zhejiang Province, 310009 Hangzhou, China. [6]Songjiang Institute and Songjiang Hospital, Shanghai Jiao Tong University School of Medicine, Shanghai, China. [7]College of Automation, Hangzhou Dianzi University, 310018 Hangzhou, China. [8]These authors contributed equally: Mengmeng Xia, Qiyue Wang, Yamin Liu, Chunyan Fang. ✉e-mail: lfy@zju.edu.cn; liuhongying@hdu.edu.cn; wangyuz@mail.ahnu.edu.cn; dsling@sjtu.edu.cn

cyano, and azide, as bioorthogonal handles to enable the desired manipulations[6–9]. By harnessing the power of bioorthogonal reactions, researchers have gained an unprecedented opportunity to achieve precise control over the dynamics of nanoparticles.

Over the past decade, various attempts have been made to manipulate the dynamic performance of nanoparticles in vivo using different bioorthogonal click reactions. Through the utilization of specially designed functional groups as bioorthogonal handles, these reactions have facilitated remarkable advancements in target-oriented conjugation, stimuli-responsive self-assembly, and controlled movement of nanoparticles[10–16]. Despite the remarkable progress achieved thus far, it is important to note that bioorthogonal click reactions have predominantly been confined to organic pericyclic reactions. These reactions often necessitate intricate designs and the laborious introduction of specific chemical groups onto the surface of nanoparticles. Consequently, their successful implementation heavily relies on the expertise of trained synthetic chemists and frequently exhibits limited efficiency in terms of nanoparticle manipulation[1,12–14,16,17]. Furthermore, in many instances, the construction of two complementary blocks is required to facilitate the bioorthogonal reaction, adding complexity to the overall system[18,19]. Thus, there is a pressing need to extend the scope of these reactions to encompass in vivo manipulation of the dynamic performance of nanoparticles, as this expansion holds fundamental significance for the field's advancement.

The dimerization of tyrosine (Tyr) is a naturally occurring and biocompatible free radical reaction[20–22]. Its inherent advantages, including the use of naturally existing handles without the need for synthetically developed functional groups or complex molecular designs, position it as a promising candidate for the in vivo manipulation of nanoparticles. The application of dityrosine cross-linking has already been demonstrated in various fields, such as cellular imaging, material synthesis, and photocatalysis[23–27]. However, utilizing this reaction for in vivo nanoparticles manipulation has not been previously reported, likely due to the specific reaction conditions required, such as UV light irradiation[28–30], enzyme[31], or strong base conditions[32,33]. Enabling such a reaction to occur in living organisms with sufficient controllability is challenging. Considering the intrinsic optical, thermal, and catalytic properties of functional nanoparticles, we propose that integrating nanoparticle catalysis with the Tyr dimerization reaction may provide a practical approach for dynamically manipulating nanoparticles in vivo.

In this study, we present a facile approach for manipulating the assembly of nanoparticles in vivo through a self-catalysis-instructed dimerization of Tyr reaction (Fig. 1). Our model system involves the engineering of ultrasmall copper sulfide nanoparticles (CuS NPs) with defects using an iodine (I) doping strategy, resulting in I-doped CuS NPs (CuS-I NPs) that exhibit excellent catalytic performance for Tyr dimerization. By leveraging self-catalysis-regulated Tyr dimerization,

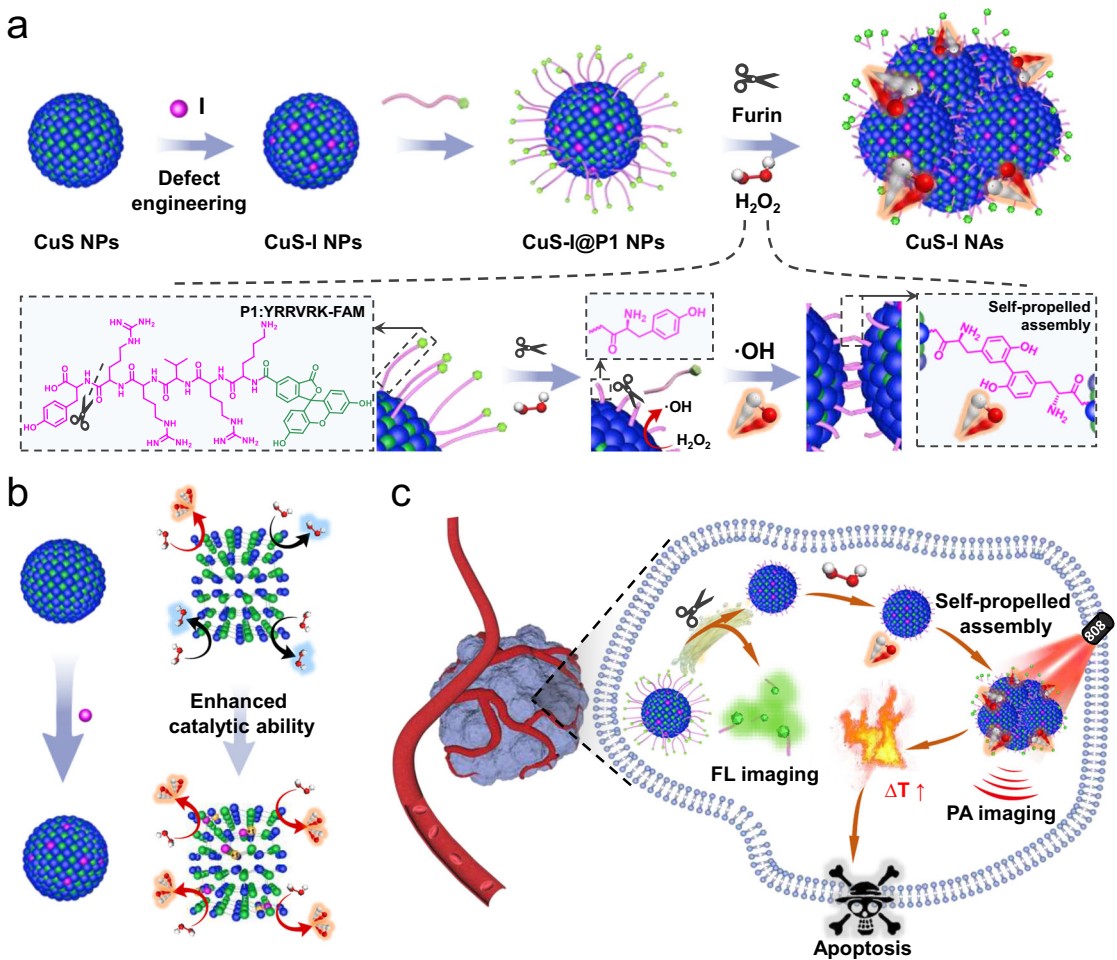

**Fig. 1 | Schematic illustration of the design and self-propelled in-situ formation of CuS-I nano-assemblies (NAs) via Tyr-Tyr condensation reactions catalyzed by Tyr-modified peptide-capped I-doped CuS nanoparticles (CuS-I@P1 NPs).** **a** Schematic illustration of the synthetic procedures of CuS-I@P1 NPs, and the self-propelled assembly through the hydroxyl radical (•OH) produced by self-catalytic reaction of CuS-I@P1 NPs in response to furin and hydrogen peroxide ($H_2O_2$). **b** Schematic illustration of the enhanced catalytic ability of CuS-I NPs in generating •OH compared to CuS NPs. **c** Schematic illustration of the specifically activated imaging and therapeutic functionalities of CuS-I NPs via self-propelled assembly triggered by overexpressed furin and $H_2O_2$.

we achieve precise control over the dynamic performance of nanoparticles in vivo. Specifically, we introduce Tyr-containing peptides (P1) with furin-responsiveness as ligands onto the surface of CuS-I NPs, which enables controllable and targeted assembly of CuS-I NPs triggered by the overexpressed furin in tumour cells, facilitating specifically activated photothermal effects, photoacoustic (PA) imaging and fluorescence imaging. The self-propelled nanoparticle assembly via Tyr-Tyr condensation reactions offers several key advantages: (1) The synthesis route is facile and cost-effective, devoid of complex chemical reaction sites. (2) The employed self-catalyzed free radical reaction of CuS-I NPs demonstrates high specificity and efficiency, thereby minimizing undesirable side reactions. This in vivo manipulation of

nanoparticles, utilizing natural handles in conjunction with the catalytic properties of nanoparticles, opens up new prospects for target-activated, smart nanomedicine.

## Results

### Tyrosine dimerization induced by CuS-I nanoparticles with enhanced catalytic performance

To promote the dimerization of Tyr, which relies on the generation of •OH[23], we utilized an I-doping strategy to enhance the catalytic performance of CuS NPs. First, monodispersed ultrasmall CuS-I NPs with a hydrodynamic diameter of ~ 10 nm are prepared through a facile one-pot hydrothermal reaction (Fig. 2a and Supplementary Fig. 1). Energy-

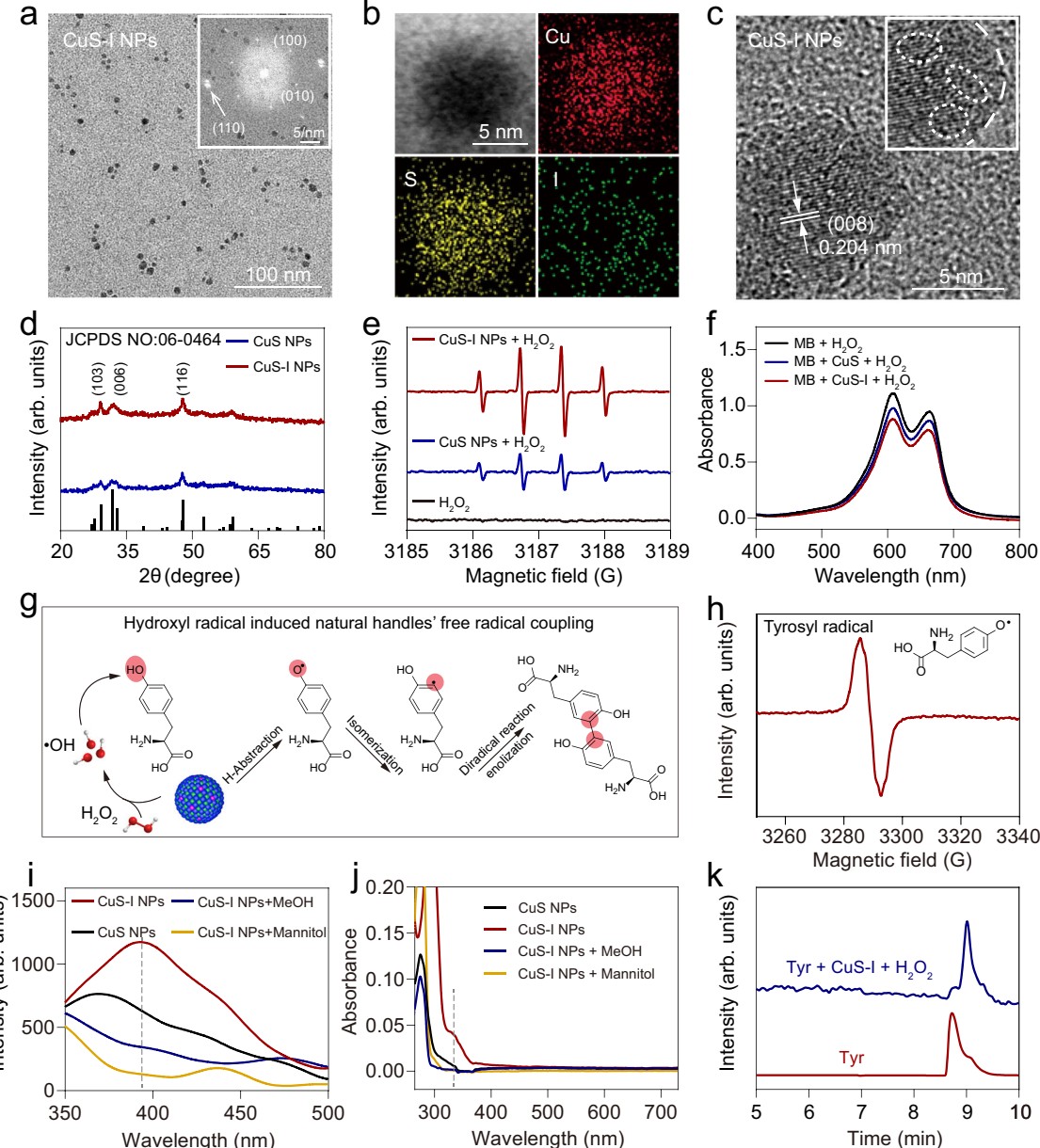

**Fig. 2 | Characterization and catalytic performance of CuS-I NPs. a** Transmission electron microscopy (TEM) image of CuS-I NPs with the corresponding SAED pattern shown as the inset. **b** EDX elemental mapping of CuS-I NPs. **c** HRTEM image of CuS-I NPs, with vacancy defects marked by the white circles. **d** XRD spectrum of CuS NPs and CuS-I NPs. **e** EPR spectrum of 5, 5-dimethyl-1-pyrroline-N-oxide (DMPO)-•OH spin adducts generated by CuS-I NPs and CuS NPs in the presence of DMPO as the •OH trapping agent. **f** Ultraviolet visible (UV-Vis) absorbance spectrum of MB incubated with CuS NPs and CuS-I NPs in the

presence of H₂O₂. **g** Mechanism of the Tyr dimerization catalyzed by CuS-I NPs under a mild condition mimicking tumour intracellular microenvironment (phosphate buffer solution (PBS) buffer containing 100 µM H₂O₂, and pH = 6.5). **h** EPR spectrum of DMPO-tyrosyl radical spin adducts when Tyr treated with CuS-I NPs in the presence of H₂O₂. **i, j** Fluorescence emission curves (**i**) and UV-Vis absorbance spectrum (**j**) of Tyr + H₂O₂ after different treatments. **k** High-performance liquid chromatography (HPLC) trace of Tyr and Tyr treated with CuS-I NPs in the presence of H₂O₂.

dispersive X-ray (EDX) results confirm that Cu, S, and I are homogeneously distributed in CuS-I NPs (Fig. 2b and Supplementary Fig. 2), demonstrating the successful introduction of I atoms. The I: (I + S) elemental ratio is quantified to be ~1.14% by using inductively coupled plasma mass spectrometry (Supplementary Table 1). The selected area electron diffraction pattern (SAED) of CuS-I NPs displays a hexagonal close-packed structure (Fig. 2a, inset), indicating the high degree of crystallinity after I-doping, which is also confirmed by high-resolution transmission electron microscopy (HRTEM) and X-ray diffraction (XRD) (Fig. 2c, d). Compared with CuS NPs (Supplementary Fig. 3), CuS-I NPs show a higher density of S defects, which can enhance their catalytic performance by increasing the active sites on the surface (Fig. 2c)[34,35]. This observation aligns with the electron paramagnetic resonance (EPR) pattern, which reveals more pronounced S vacancies in CuS-I NPs[36] (Supplementary Fig. 4). EPR spectroscopy exhibits a more obvious •OH signal for CuS-I NPs compared with CuS NPs

(Fig. 2e). In the presence of $H_2O_2$, the absorbance variation of methylene blue (MB) at 664 nm[37] mixed with CuS-I NPs is 2.03-fold lower than that of CuS NPs, demonstrating that I-doping significantly enhances the Fenton-like reaction to produce •OH (Fig. 2f).

Based on the superior catalytic performance of CuS-I NPs, we further explore their potential in catalyzing the dimerization of natural handle Tyr. The dimerization mechanism of Tyr catalyzed by CuS-I NPs is depicted in Fig. 2g. In this process, the •OH generated by CuS-I NPs captures a hydrogen atom from the phenolic hydroxyl group, resulting in the formation of tyrosyl radical (Fig. 2h). Notably, the typical fluorescence emission and UV-Vis absorption associated with dityrosine can only be observed when incubated Tyr with CuS-I NPs (Fig. 2i, j), demonstrating the capability of CuS-I NPs to induce Tyr dimerization, which is also confirmed by HPLC result (Fig. 2k). Addition of reactive oxygen quenching agents (methanol or mannitol) significantly inhibits the formation of dityrosine (Fig. 2i, j),

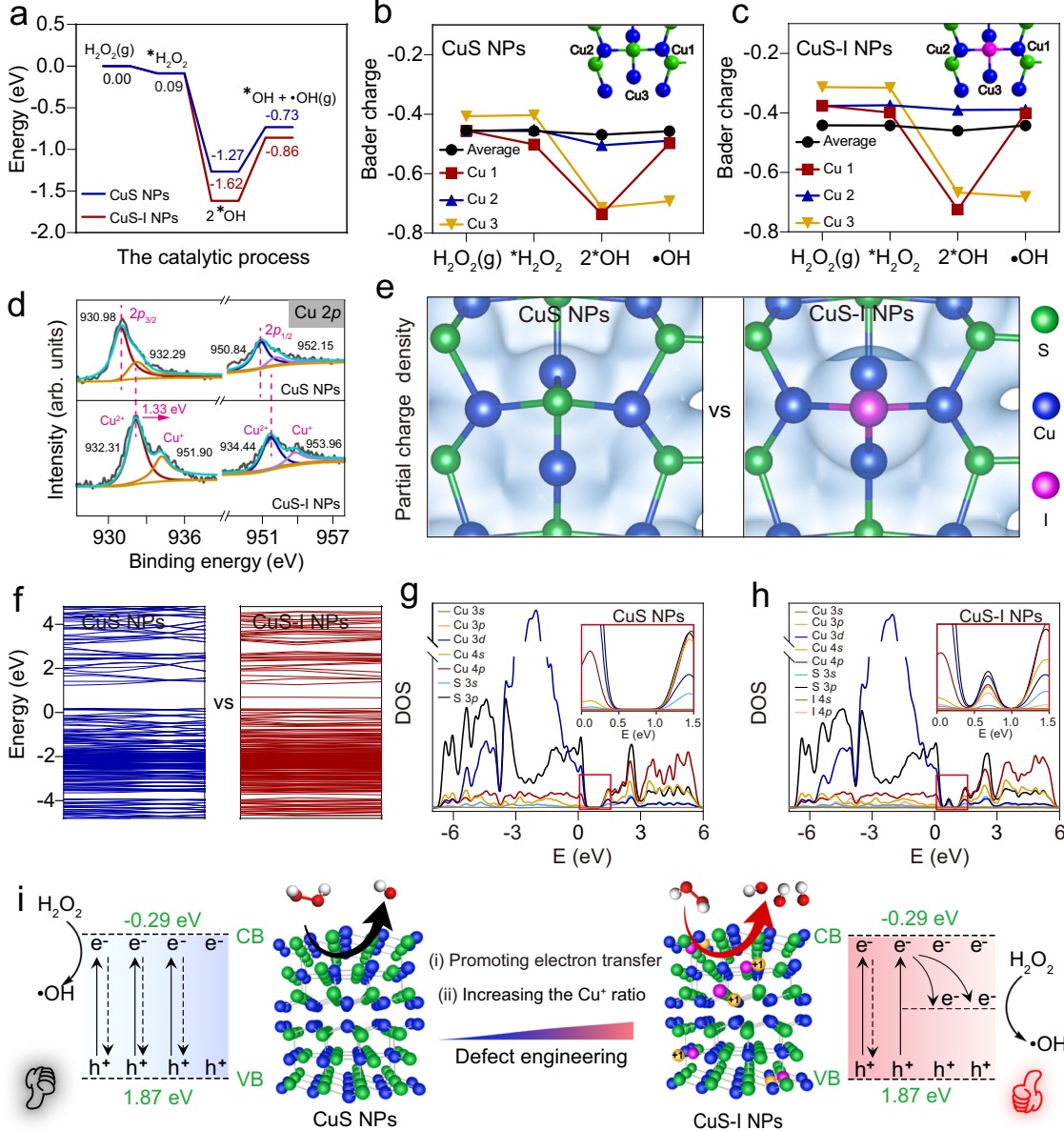

**Fig. 3 | Mechanism for the enhanced catalytic performance of CuS-I NPs.**
**a** Reaction energy profiles depicting the Fenton-like catalysis of CuS-I NPs and CuS NPs. **b, c** The Bader charge analysis of Cu atoms in CuS NPs (**b**) and CuS-I NPs (**c**). **d** High-resolution XPS of Cu 2*p*. **e** Charge density analysis of CuS and CuS-I NPs.

**f** Energy band diagram of CuS NPs and CuS-I NPs. **g, h** Partial density of states (PDOS) of CuS NPs (**g**) and CuS-I NPs (**h**). **i** Schematic illustration of mechanism behind the enhanced catalytic performance of CuS-I NPs compared to that of CuS NPs.

suggesting the indispensability of •OH generated by CuS-I NPs in Tyr dimerization.

## Mechanistic studies on the catalytic performance of CuS-I nanoparticles

Furthermore, we investigate the catalytic mechanism of CuS-I NPs in •OH generation by X-ray photoelectron spectroscopy (XPS) and density functional theory (DFT) calculations. It is observed that $H_2O_2$ molecules on the surface of CuS-I NPs are more easily decomposed due to the lower decomposition energy (−1.53 eV) than that of CuS NPs (−1.18 eV) (Fig. 3a). Upon analyzing the Bader charge of Cu atoms around the I-doping sites (Fig. 3b, c), it is found that the charge of Cu atoms near the I-doping sites increases compared to that before doping, and is higher than the average charge of Cu atoms on the surface. This observation indicates that Cu atoms near the I-doping sites tend to be reduced from bivalent to monovalent, resulting in the improved catalytic activity, which is also confirmed by XPS results. As shown in Fig. 3d, the surface $Cu^{2+}/Cu^{+}$ ratio decreases from 1.46 to 0.92 after I-doping. Moreover, there is a positive displacement of the Cu orbital and a negative displacement of the S orbital after I-doping (Fig. 3d and Supplementary Fig. 5), indicating the enhanced electronic interaction between Cu atoms and S atoms, which has a favorable effect on the self-reconstruction process of the surface during catalysis[38,39]. Charge density analysis proves that the doped I atom has a higher electron density compared with the S atom at the same site (Fig. 3e), making it easier for the $Cu^{+}$ generation. Furthermore, the PDOS (Fig. 3f–h) reveals the emergence of a new state density peak for CuS-I NPs, which leads to a reduced electron transportation barrier, thus facilitating the completion of the catalytic reaction cycle[40–42]. Overall, I-doping strategy enhances the Fenton-like catalytic activity of CuS-I NPs by increasing the $Cu^{+}$ ratio and promoting electron transfer on the surface of the nanoparticles (Fig. 3i).

## Enzyme-instructed self-propelled assembly of CuS-I@P1 NPs

Encouraged by the promising results of CuS-I NPs in catalyzing the dimerization of Tyr, we hypothesize that the integration of nanoparticle catalysis with the Tyr dimerization reaction enables the dynamic manipulation of nanoparticles. To proof of concept, we modified CuS-I NPs with a furin-responsive short peptide (5-Carboxyfluorescein-Lys-Arg-Val-Arg-Arg-Tyr-OH, FAM-KRVRR ↓ Y, termed P1; ↓ indicates the cleavage site of furin) connected to Tyr and labeled with FAM via a facile condensation reaction to obtain CuS-I@P1 NPs[43]. Fourier transform infrared spectrum (FT-IR) and UV-Vis spectrum demonstrate the successful modification of P1 on the surface of CuS-I NPs (Supplementary Fig. 6). The CuS-I@P1 NPs remain stable at room temperature for up to 7 days without noticeable size change, indicating their good colloidal stability (Supplementary Fig. 7). As a trans-Golgi protein convertase, furin is upregulated in multiple malignant tumours and has emerged as an important biomarker for subcellular organelle-targeting theranostics[44–46]. Upon internalization into tumour cells, the intracellular furin cleaves the RVRR fragment of CuS-I@P1 NPs, resulting in the exposure of Tyr and release of fluorophore FAM-loaded peptide fragment. Furthermore, the overexpression of $H_2O_2$ in tumour cells activates the Fenton-like reaction capability of CuS-I NPs, which leads to the generation of •OH, thereby facilitating Tyr dimerization and the self-propelled in situ assembly of CuS-I NPs into CuS-I NAs (Fig. 4a). For comparison, CuS@P1 NPs and CuS-I NPs modified with a scrambled peptide (FAM-Lys-Arg-Lys-Arg-Ala-Tyr-OH, FAM-KRKRAY, P1-scr) that cannot be cleaved by furin were also prepared (CuS-I@P1-scr NPs) (Supplementary Fig. 8). Furthermore, we optimized the coupling of Tyr on CuS-I NPs by introducing varying quantities of Tyr. It was observed that as the Tyr feeding amount increased, the assembly size of CuS-I NPs exhibited a gradual initial increase followed by stabilization (Supplementary Fig. 9). After treatment with furin and $H_2O_2$, the average hydrodynamic diameter of CuS-I@P1

increases from -14.17 nm to -79.85 nm (Fig. 4b and Supplementary Fig. 10), and a significant fluorescence characteristic peak of dityrosine is detected, in stark contrast to the CuS@P1 NPs and CuS-I@P1-scr NPs (Fig. 4c). Moreover, the characteristic peak of dityrosine is not observed when treated with other enzymes (Fig. 4d), indicating the superior specificity of CuS-I@P1 NPs in response to furin.

The FAM fluorescence emission can be quenched due to fluorescence resonance energy transfer (FRET)[47–49] between FAM group and CuS-I NPs (Fig. 4e and Supplementary Fig. 11). After incubation with furin, the fluorescence intensity of the CuS-I@P1 solution at 518 nm increases by approximately 3.5-fold, while no noticeable change is observed in the CuS-I@P1-scr NPs-treated group (Fig. 4f, g and Supplementary Fig. 12). Additionally, the peptide content was quantified to be 56.90% based on the peptide P1 standard curve (Supplementary Fig. 13). Notably, I-doping not only enhances the catalytic activity of CuS NPs but also augments their photothermal conversion effect (Supplementary Fig. 14). Moreover, the self-propelled assembly of CuS-I NPs can further improve the photothermal effect. The light absorption of CuS-I NAs at 808 nm was enhanced compared to that of CuS-I NPs (Supplementary Fig. 15). Upon exposure to an 808 nm laser at 2 W cm$^{-2}$ for 5 min, the temperature of CuS-I NAs increases by 29.2 °C, exhibiting a close to twofold temperature elevation compared to CuS-I@P1 NPs (Fig. 4h, i). The improved photothermal effect of CuS-I NAs can be attributed to several potential mechanisms. On the one hand, the alignment of CuS-I NAs with photonic-crystal microstructures enhances light absorption[50]. On the other hand, the assemblies exert substantial influence on the electronic structures of nanomaterials, resulting in a significant improvement in the molar extinction coefficient within the NIR region[51]. As shown in Supplementary Fig. 16, CuS-I NAs exhibited a bandgap of 1.81 eV, which is lower than the 2.03 eV observed for CuS-I NPs, resulting in an augmented molar extinction coefficient and improved light absorption at 808 nm in comparison to dispersed CuS-I NPs[52]. These collective results demonstrate that Tyr-containing enzyme-responsible peptides allow for the manipulation of targeted assemblies of CuS-I@P1 NPs through the self-catalysis-regulated natural handle Tyr dimerization, enabling specific activation of imaging and therapeutic effects.

## Self-propelled assembly of CuS-I@P1 NPs at cellular level

The self-propelled assembly of CuS-I@P1 NPs in cancer cells was investigated using furin-overexpressing human breast adenocarcinoma epithelial cells (MDA-MB-468 cells). To verify intracellular furin-instructed FAM release, co-localization of CuS-I@P1 NPs and Golgi in the MDA-MB-468 cells was examined. The green fluorescence from FAM merges well with the red fluorescence from Golgi Tracker, providing evidence for the furin-responsive behavior of CuS-I@P1 NPs in living cells (Fig. 5a, b and Supplementary Fig. 17). Moreover, the CuS-I@P1 NPs-treated group exhibits a significantly higher reactive oxygen species (ROS) generation in MDA-MB-468 cells than the CuS-I@P1-scr NPs-treated group (Fig. 5c), thus promoting the self-propelled nanoparticle aggregation within living cells. TEM observation revealed the assembly of CuS-I@P1 NPs in the lysate of MDA-MB-468 cells, along with the typical fluorescence emission at -410 nm associated with dityrosine (Supplementary Fig. 18). Also, we observed large areas of assemblies within cells treated with CuS-I@P1 NPs (Supplementary Fig. 19a), while only a few CuS-I@P1-scr NPs remained in the cytoplasm of CuS-I@P1-scr-treated cells (Supplementary Fig. 19b). These results further verify that the assembly of CuS-I@P1 NPs in tumour cells is achieved through self-catalysis-regulated Tyr dimerization. To confirm the role of furin in this process, we incubated MDA-MB-468 cells pretreated with furin inhibitor (Supplementary Fig. 19c), and normal 293T cells (Supplementary Fig. 19d) with CuS-I@P1 NPs. Interestingly, the nanoparticles remain relatively dispersed in the cytoplasm due to the absence of furin (Supplementary Fig. 19c, d), indicating the indispensable role of furin in guiding the self-propelled intracellular

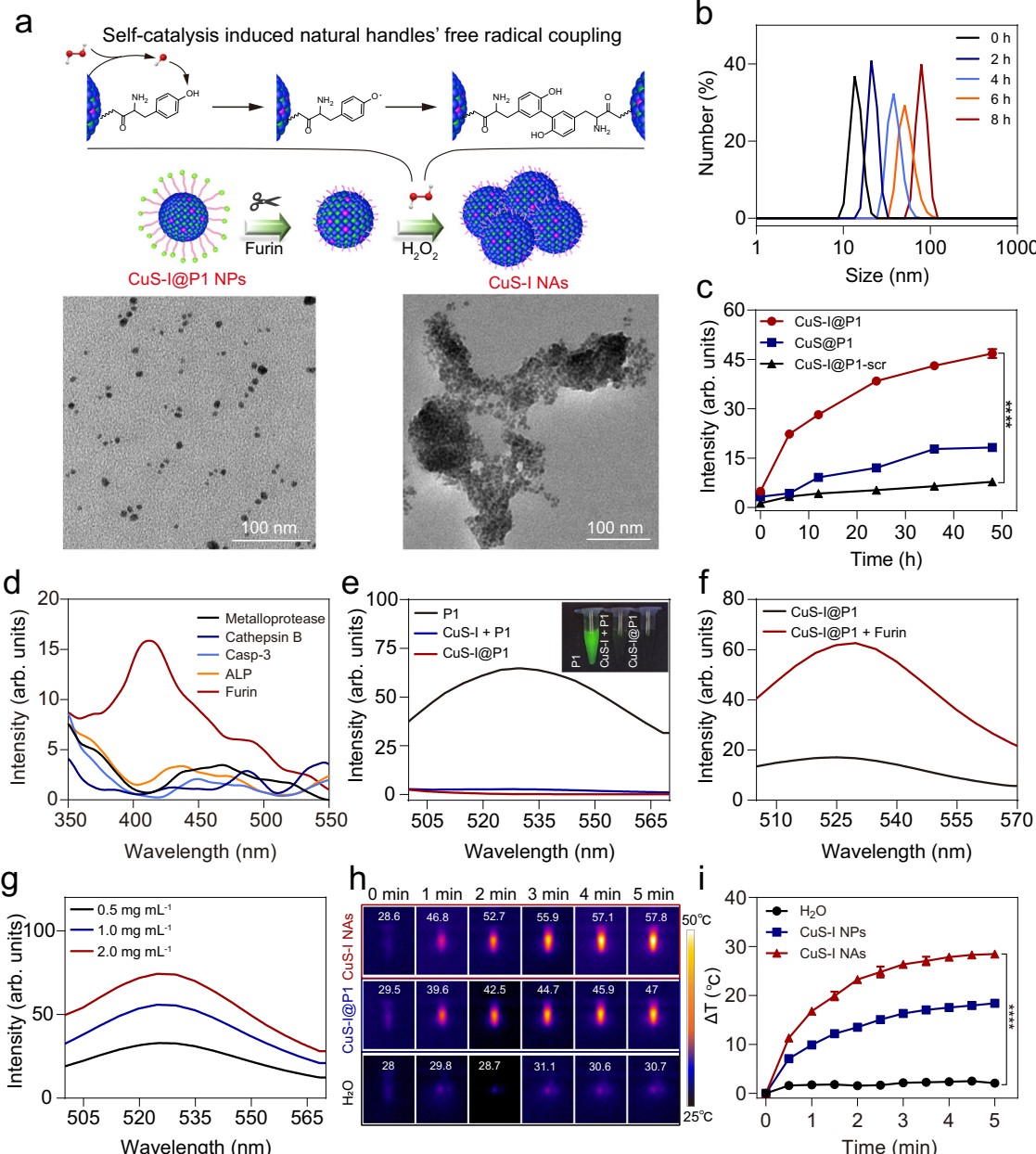

**Fig. 4 | Design and characterization of self-catalysis-instructed self-propelled assembly of CuS-I@P1 NPs. a** Scheme of the self-propelled in situ assembly of CuS-I NPs into CuS-I NAs and TEM images of CuS-I@P1 NPs before (left) and after (right) treated with furin and $H_2O_2$ for 8 h. **b** Hydrodynamic size distribution of CuS-I@P1 NPs incubated with furin and $H_2O_2$ for 0, 2, 4, 6, and 8 h. **c** Fluorescence signals of dityrosine ($\lambda_{ex} = 315$ nm) after CuS-I@P1, CuS@P1 and CuS-I@P1-scr incubated with furin + $H_2O_2$ for different times. ****$P < 0.0001$. All data are presented as means ± SEM, $n = 3$ independent experiments. Statistical significance was analyzed by unpaired, two-tailed Student's $t$ tests. **d** Fluorescence intensities of dityrosine ($\lambda_{ex} = 315$ nm) after CuS-I@P1 incubated with metalloprotease, Cathepsin B, Caspase-3 (Casp-3), alkaline phosphatase (ALP) and furin in the presence of $H_2O_2$.

**e** Fluorescence emission spectrum of FAM under the same peptide concentration (3 μg mL$^{-1}$) ($\lambda_{ex} = 490$ nm). **f** Fluorescence emission spectrum of FAM in CuS-I@P1 NPs solution before and after treatment with furin. **g** Fluorescence emission spectrum of FAM in CuS-I@P1 NPs solution with different peptide concentrations after treatment with furin. **h** Infrared thermal images of $H_2O$, CuS-I NPs, and CuS-I NAs solution exposed to laser irradiation for 5 min (808 nm, 2 W cm$^{-2}$). **i** Temperature change curves of $H_2O$, CuS-I NPs and CuS-I NAs solution with laser irradiation for 5 min (808 nm, 2 W cm$^{-2}$). ****$P < 0.0001$. All data are presented as means ± SEM, $n = 3$ independent experiments. Statistical significance was analyzed by unpaired, two-tailed Student's $t$ tests.

assembly of CuS-I@P1. CuS-I@P1 NPs show a dose-dependent cytotoxicity on MDA-MB-468 cells under NIR laser irradiation (Fig. 5d, e), while exhibiting negligible cytotoxicity towards furin-negative 293T cells (Fig. 5f and Supplementary Fig. 20), demonstrating the highly specific tumour therapeutic potential of CuS-I@P1 NPs with natural Tyr handle. The live/dead cell staining assay further confirms the remarkable cell damage induced by CuS-I@P1 NPs under laser irradiation, in sharp contrast to CuS-I@P1-scr NPs (Fig. 5g).

## Self-propelled assembly of CuS-I@P1 NPs for tumour-specific imaging and therapy

We further investigate the self-propelled assembly of CuS-I@P1 NPs for in vivo tumour imaging and therapy. The dual-modal imaging capability of CuS-I@P1 NPs can be selectively triggered by overexpressed furin and $H_2O_2$ in tumour cells (Fig. 6a). A significant fluorescence signal is detected in the tumour region of CuS-I@P1-treated mice, and the signal remains consistently high for more than 1 h, indicating the

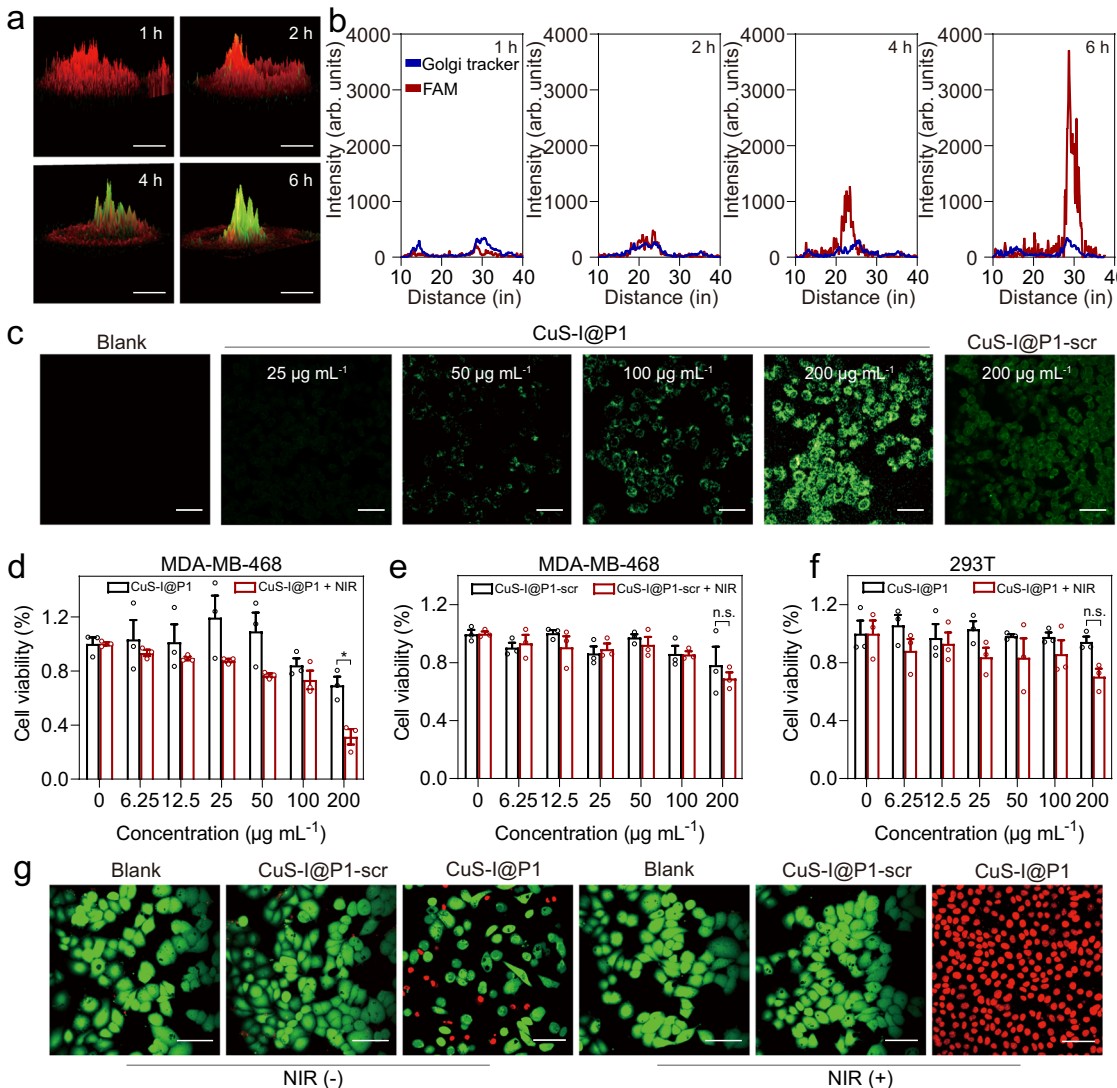

**Fig. 5 | Enzyme-instructed self-propelled assembly of CuS-I@P1 NPs in cancer cells. a** Confocal fluorescence images of MDA-MB-468 cells stained with Golgi Tracker after co-incubation with CuS-I@P1 NPs at different time points, (red indicates Golgi Tracker, green indicates FAM) Scale bar: 10 μm. **b** Co-localization of the fluorescence curves at different time points. **c** Confocal laser scanning microscope images of MDA-MB-468 cells stained with 2′,7′-Dichlorodihydrofluorescein diacetate (DCFH-DA) after incubation with different concentrations of CuS-I@P1 NPs or CuS-I@P1-scr NPs for 8 h. Scale bar: 50 μm. **d, e** Cell viability of MDA-MB-468 cells incubated with different concentrations (0–200 μg mL⁻¹) of CuS-I@P1 (**d**) or CuS-I@P1-scr (**e**) for 8 h with (or without) NIR irradiation. *P = 0.0109 (**d**); P = 0.5225 (**e**). All data are presented as means ± SEM, n = 3 independent experiments. Statistical significance was analyzed by unpaired, two-tailed Student's t tests (**d–f**). n.s. no significant difference. **f** Cell viability of 293T cells incubated with different concentrations (0-200 μg mL⁻¹) of CuS-I@P1 for 8 h with (or without) NIR irradiation. P = 0.1167. All data are presented as means ± SEM, n = 3 independent experiments. Statistical significance was analyzed by unpaired, two-tailed Student's t tests (**d–f**). **g** Confocal fluorescence images of MDA-MB-468 cells co-stained with calcein AM (green, live cells) and propidium iodide (red, dead cells) after treatment with PBS, CuS-I@P1, CuS-I@P1-scr with or without NIR laser irradiation. Scale bar: 50 μm.

efficient furin-responsive property of CuS-I@P1 at tumour sites (Fig. 6b, c). As photothermal effect is accompanied by PA signals[14], real-time PA imaging was also performed. Figure 6d, e illustrates a significantly stronger increase in the PA signal at the tumour site of CuS-I@P1-treated mice compared to the other two groups, which is likely attributed to the intracellular assembly of CuS-I@P1 NPs induced by furin and H₂O₂.

MDA-MB-468 tumour-bearing mice were used to evaluate the potential of CuS-I@P1 NPs for tumour photothermal therapy. In principle, CuS-I@P1 NPs undergo a self-catalytic reaction in response to furin and H₂O₂, leading to the self-assembly of nanoparticles (Fig. 7a). However, CuS@P1 NPs are unable to self-assemble due to their unsatisfactory catalytic activity, while CuS-I@P1-scr fails to self-assemble because of surface modification rendering the fragment unrecognizable by furin (Fig. 7a). We observed that the temperature

increasing at the tumour site after 5 min of irradiation was significantly higher at 8 and 16 h compared to 1, 2, or 4 h post-injection of CuS@P1 NPs. Due to the minimal difference in temperature increase between 8 and 16 h, we chose to perform PTT on tumours at 8 h post administration (Supplementary Figs. 21–23). As shown in Fig. 7b, c, the temperature at the tumour site of the mice treated with CuS-I@P1 NPs increased to 48 °C in 5 min, which was significantly higher than that of the CuS-I@P1-scr group (41.5 °C), CuS@P1 group (40.9 °C), and PBS group (36.9 °C). Moreover, a significant amount of nano-assemblies were observed within the tumour sections of the CuS-I@P1 NPs-treated group, but were not found in the CuS-I@P1-scr or CuS@P1 NPs-treated groups (Supplementary Figs. 24 and 25). These results suggest that the self-catalysis-regulated intracellular assembly of CuS-I NPs enables superior photothermal treatment in vivo.

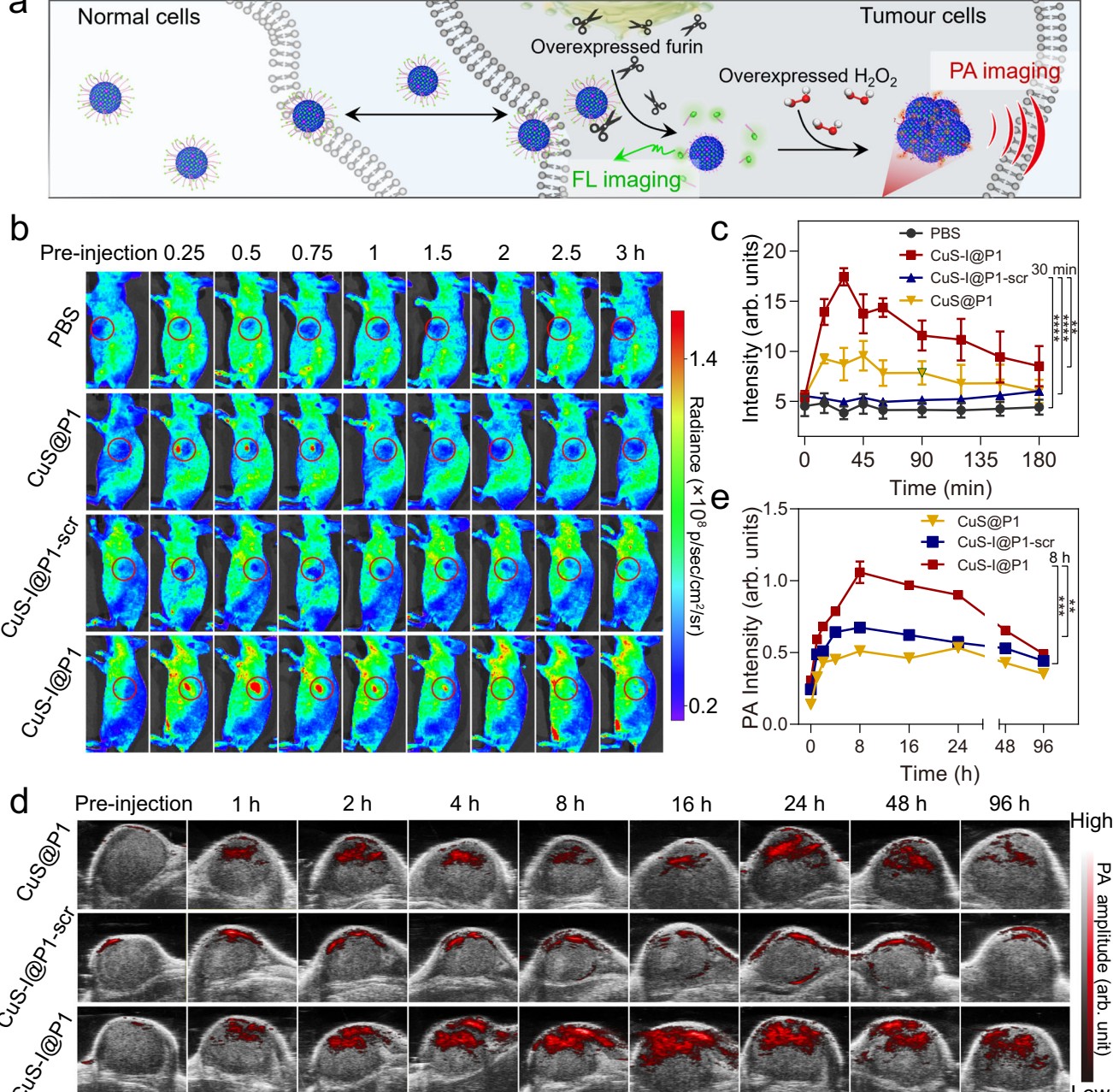

**Fig. 6 | Self-propelled assembly of CuS-I@P1 NPs for tumour-specific imaging.**
**a** Schematic illustration of the self-propelled assembly of CuS-I@P1 NPs in vivo for tumour-specific dual-modal imaging. **b** The time-dependent fluorescence images of MDA-MB-468 tumour-bearing mice after intratumour administration of PBS, CuS@P1, CuS-I@P1 and CuS-I@P1-scr. (The circles represent the locations of tumours). **c** The time-dependent quantitative calculation of the average fluorescence intensity at the tumour area. ****$P < 0.0001$; **$P = 0.0325$. All data are presented as means ± SEM, $n = 3$ independent experiments. Statistical significance was analyzed by unpaired, two-tailed Student's $t$ tests. **d** The PA imaging of MDA-MB-468-tumour-bearing mice with different treatment before and after intratumour administration. **e** The average PA intensity increment at 830 nm ($\Delta PA_{average}$) of tumour areas after intratumour administration of CuS@P1, CuS-I@P1 and CuS-I@P1-scr. ***$P = 0.0004$, **$P = 0.0024$. All data are presented as means ± SEM, $n = 3$ independent experiments. Statistical significance was analyzed by unpaired, two-tailed Student's $t$ tests.

Tumour volume and mouse weight were monitored for 36 days, and the CuS-I@P1 + NIR group showed the highest tumour inhibition rate, demonstrating the remarkable antitumour performance of CuS-I NPs due to self-catalysis-regulated assembly (Fig. 7d). The therapeutic outcome was also verified by histomorphology analysis. Tumour tissues were harvested on day 36 and subjected to hematoxylin-eosin (H&E) staining, revealing the noticeable discrete cancer cells and tumour shrinkage in the CuS-I@P1 + NIR group (Fig. 7e). Furthermore, the mice showed a steady increase in body weight throughout the treatment (Fig. 7f). Additionally, no apparent abnormalities were observed in H&E-stained slices of major organs, serum biochemistry parameters, and routine blood parameters, indicating the excellent biocompatibility of the combination therapy (Supplementary Figs. 26 and 27). Collectively, all this evidence strongly supports that the self-propelled assembly of CuS-I@P1 NPs by means of dityrosine cross-linking, without the need for tedious synthesis, may offer a useful tool for effective cancer treatment.

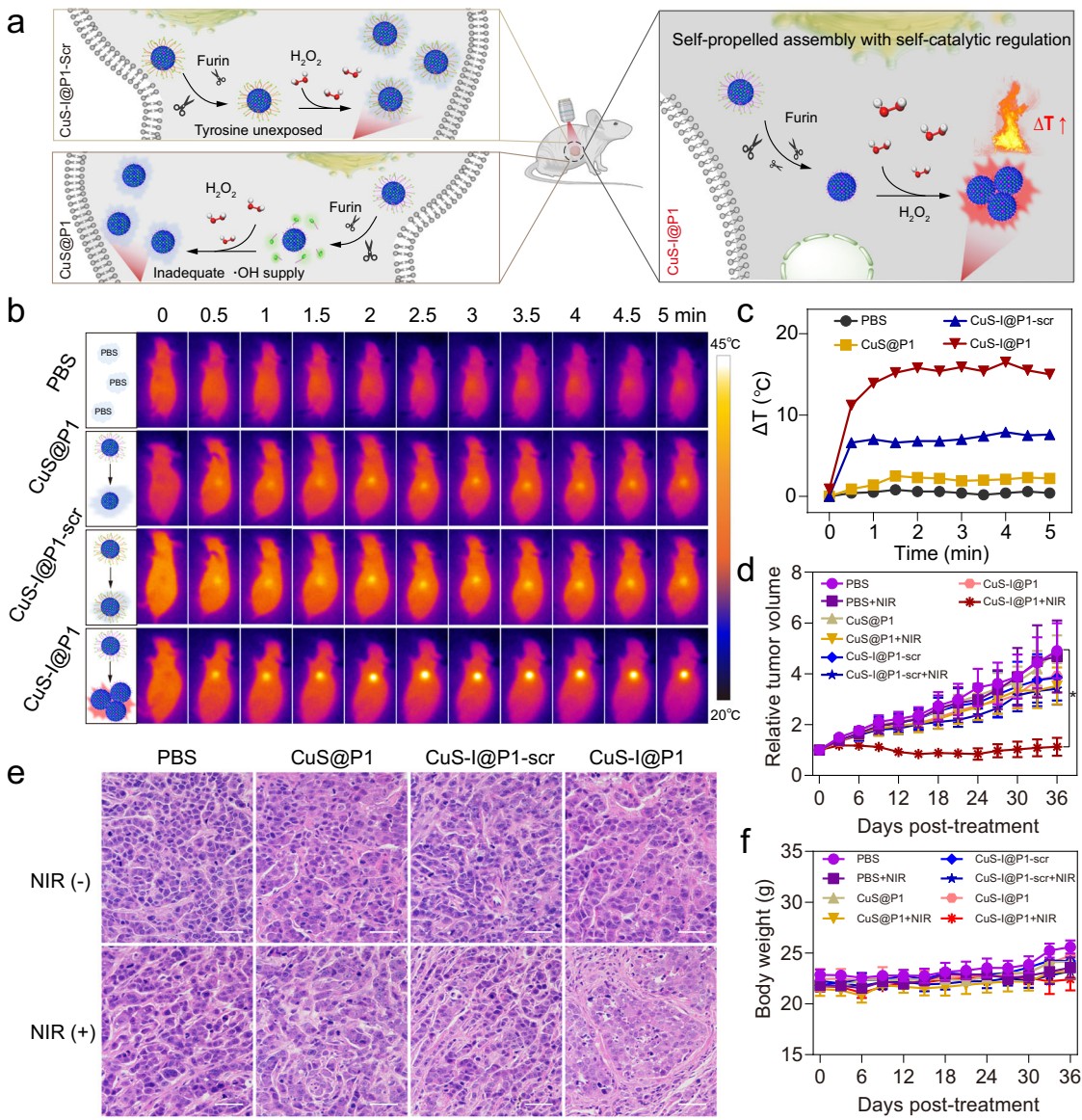

**Fig. 7 | Antitumour efficacy of self-propelled assembly of CuS-I@P1 NPs in vivo. a** Schematic illustration of tumour-bearing mice for different treatments. **b** Infrared thermal images of tumour-bearing mice with different treatments under laser irradiation (808 nm, 2 W cm$^{-2}$, 5 min). **c** Temperature changes at the tumour sites of tumour-bearing mice subjected to laser irradiation (808 nm, 2 W cm$^{-2}$, 5 min) after intratumoural administration of PBS, CuS-I@P1 and CuS-I@P1-scr for 8 h. **d** Relative tumour volume curves of MDA-MB-468 tumour-bearing mice under distinct treatments for 36 d. $^*P = 0.0223$. All data are presented as means ± SEM, $n = 5$ independent experiments. Statistical significance was analyzed by unpaired, two-tailed Student's $t$ tests. **e** H&E staining images of tumours from different groups on day 36. Scale bar: 50 μm. **f** Body weight curves of MDA-MB-468-tumour-bearing mice under distinct treatments for 36 d.

## Discussion

In summary, we have developed a nanoparticle self-catalysis-regulated in situ assembly approach to achieve dynamic manipulation of nano-particles in vivo without complicated design and tedious synthesis. By introducing Tyr-modified enzyme-responsive peptides as ligands on the surface of I-doped CuS NPs, we have achieved controllable, self-propelled assembly of CuS-I NPs via Tyr-Tyr condensation reactions catalyzed by the nanoparticles themselves inside tumour cells. In vitro and in vivo experiments have demonstrated the tumour-specific furin-instructed intracellular formation of CuS-I nano-assemblies through dityrosine bonds, thus exhibiting activatable dual-modal imaging capability and enhanced photothermal effect.

Such an integrative strategy using natural handles and the cata-lytic properties of nanoparticles provides an efficient and general-izable avenue for tumour-specific imaging and therapy. Our approach offers significant advantages over current strategies by avoiding

complicated designs and tedious synthesis, thus paving the way for the development of more effective nanotherapeutics with precise and targeted control over assembly and functional activation.

## Methods

### Materials

Sodium sulfide nonahydrate (Na$_2$S·9H$_2$O), copper chloride dehydrate (CuCl$_2$·2H$_2$O), sodium iodide (NaI), poly-(vinylpyrrolidone) (PVP) K$_{15}$ (viscosity average molecular 10,000 Da), Tyr, MB, 1-ethyl-3-(3-dime-thylaminopropyl) carbodiimide hydrochloride (EDC HCl), N-hydroxy succinimide (NHS), N-2-Hydroxyethylpiperazine-N-2-Ethane sulfonic acid (HEPES), methyl alcohol (MeOH), trifluoroacetic acid (TFA) were obtained from Aladdin Reagent Co., Ltd. (Shanghai, China), furin was purchased from Biolabs (one unit (U) corresponds to the amount of furin that releases 1 pmol of methylcoumarinamide (MCA) from the fluorogenic peptide Boc-RVRRMCA (Bachem) in 1 min at 30 °C).

2-aminoethanethiol, $H_2O_2$, 2,7-dichlorofluorescin diacetate and paraformaldehyde (4%) were purchased from Bidepharm Technology Co., Ltd. (Shanghai, China). Calcein-AM/PI double stain kit, 3-(4,5)-dimethylthiahiazo(-z-y1)-2,5-di- phenytetrazoliumromide (MTT) was obtained from Beyotime Biotechnology Co., Ltd. (Shanghai, China). Dulbecco's modified Eagle's medium (DMEM), fetal bovine serum (FBS), 1% penicillin-streptomycin were obtained from Thermo Fisher Scientific Inc.

## Characterization

The transmission electron microscopy images were conducted with a Hitachi HT7700 (Hitachi, Japan) field-emission transmission electron microscope under 120 kV accelerating voltage. energy-dispersive spectroscopy (Hitachi S-4800). Ultraviolet-visible absorption spectra were obtained by a U-4100 spectrometer (Hitachi, Japan). Fourier transform infrared (FT-IR-8400S) spectrometer (Shimadzu, Japan) was used for recording the FT-IR spectrum. The zeta potential analyses were performed on a Malvern Zeta sizer Nano-ZS ZEM3600 (U.K.). High-performance liquid chromatography analyses were recorded on the Shimadzu chromatographic system (Tokyo, Japan). The laser used in this study was provided from the multimode fiber coupled infrared semiconductor diode laser (808 nm, Leirui Optoelectronics Co., Ltd., Changchun, China). The chemical compositions of the prepared catalysts were examined by an X-ray photoelectron spectrometer (Escalab 250Xi, Thermo Fisher).

## Synthesis of CuS NPs

Briefly, $Na_2S$ solution (1 M, 50 µL) was added into the solution (50 mL) containing $CuCl_2$ ($1 \times 10^{-3}$ M) and PVP $K_{15}$ (0.2 g) stirring at room temperature for 5 min. The reaction mixture was heated to 60 °C until a dark green solution was obtained. After being transferred to ice-cold water, the resulting CuS NPs were purified by ultracentrifugation (6000 ×$g$, 5 min) using an Amicon Ultra-15 centrifugal filter unit Millipore and stored at 4 °C.

## Synthesis and modification of CuS-I NPs

For the synthesis of CuS-I NPs, the reaction conditions were exactly similar to that of CuS NPs, except for the addition of NaI precursor. To obtain the amination derivative of CuS-I NPs, CuS-I NPs (1 mg) were added to the PBS solution of 2-aminoethanethiol (4 mg, 5 mL) stirring for 24 h. The resultant solution was centrifuged (10,000 ×$g$, 8 min), washed with water three times and then freeze-dried to obtain CuS-I-$NH_2$.

## Modification of CuS-I NPs with peptide

A mixture of EDC HCl, NHS and P1 or P1-scr with a molar ratio of 4:4:1 was dissolved in PBS. Then, CuS-I-$NH_2$ (0.2 mg mL$^{-1}$, 5 mL) was added into the above suspension and stirred at room temperature for 12 h. The products were washed with PBS three times to remove the unreacted P1 and other reagents, and then freeze-dried to obtain CuS-I@P1 NPs and CuS-I@P1-scr NPs, respectively. CuS@P1 NPs were obtained by a similar method.

The encapsulating efficiency of peptides was calculated by the following equation:

$$\text{loading efficiency}\,(\%) = \frac{\text{Weight of loaded P1}}{\text{Weight of devoted P1}} \times \mathbf{100}\%$$

## Hydroxyl radical generating capacity of CuS-I NPs

The MB probe was used to evaluate •OH generating capacity. Firstly, CuS-I NPs or CuS NPs solution (1 mg mL$^{-1}$, 200 µL) was added into MB solution (5 mg mL$^{-1}$, 20 µL) in the presence of $H_2O_2$ (100 µM, 780 µL), and reacted at room temperature for 8 h. Then, the UV-Vis absorption intensity of mixed solution was measured.

## Fluorescence intensity detection of dityrosine

In the presence of $H_2O_2$ (100 µM, 1 mL), the Tyr solution (1 mg mL$^{-1}$, 200 µL) was mixed with CuS (1 mg mL$^{-1}$, 800 µL), CuS-I solution (1 mg mL$^{-1}$, 800 µL), CuS-I solution (1 mg mL$^{-1}$, 800 µL) + methyl alcohol (10 µL), CuS-I solution (1 mg mL$^{-1}$, 800 µL) + mannitol (10 µL), and then the fluorescence intensity of the mixed solution was measured by fluorescence spectrophotometer ($\lambda_{ex}$ = 315 nm). In the presence of $H_2O_2$ (100 µM), CuS-I@P1, CuS-I@P1-scr, CuS@P1 (200 µg mL$^{-1}$) were mixed with furin (20 U mL$^{-1}$), then the fluorescence intensity of the mixed solution was measured at different times by fluorescence spectrophotometer ($\lambda_{ex}$ = 315 nm). In the presence of $H_2O_2$ (100 µM), CuS-I@P1 solution (200 µg mL$^{-1}$) were mixed with furin, metalloprotease, Cathepsin B, Caspase-3, ALP for 8 h, then the fluorescence intensity of the mixed solution was measured by a fluorescence spectrophotometer ($\lambda_{ex}$ = 315 nm).

## HPLC analyses

HPLC analyses were performed on an Agilent 1200 HPLC system equipped with a G1322A pump and in-line diode array UV detector using an Agilent Zorbax 300SB-C18 RP column with $CH_3CN$ (0.1% of (TFA)) and water (0.1% of TFA) as the eluent. HPLC traces of Tyr (0.1 mg mL$^{-1}$), Tyr (1 mg mL$^{-1}$, 200 µL) treated with CuS-I solution (1 mg mL$^{-1}$, 800 µL) and $H_2O_2$ (100 µM, 1 mL) for 8 h at 37 °C were detected.

## Density functional theory (DFT) calculations

The density functional theory calculations were carried out using Quantum Espresso (Version 6.7). CuS bulk structure was obtained from Materials Project (https://legacy.materialsproject.org/, Materials ID: mp-555599) and optimized. Then the CuS bulk structure was cleaved to the (1 0 0) surface modeled with the (2 × 2) periodically repeated supercell, consisting of three atomic layers, with a vacuum space of 15 Å. The CuS-I surface structure was modeled based on the CuS surface with one S atom altered as I atom. The calculations were performed using the DFT method in combination with Standard solid-state pseudopotentials (SSSP) for efficiency with Perdew–Burke–Ernzerhof (PBE) exchange-correlation functional as implemented in the Quantum Espresso package. The kinetic energy cutoff for the plane-wave-basis set was set as 55 Ry. For all geometry optimizations, the energy convergence criterion for the electronic self-consistent loop and the ionic relaxation loop were set as $1.0 \times 10^{-6}$ eV and $1.0 \times 10^{-4}$ eV, respectively. $2 \times 2 \times 1$ Monkhorst–Pack k-point mesh samplings were used during the DFT calculation process.

## Fluorescence intensity detection of FAM

CuS-I@P1 solution (200 µg mL$^{-1}$) was mixed with furin (20 U mL$^{-1}$) for 8 h in dark place, then the fluorescence intensity of the mixed solution was measured by fluorescence spectrophotometer ($\lambda_{ex}$ = 490 nm).

## Calculation of band gaps for CuS-I NPs and CuS-I NAs

We utilized the Tauc plot method to calculate the band gaps (Eg) of CuS-I NPs and CuS-I NAs.

$$\text{Tauc plot}: (\alpha h\nu)^{1/n} = B(h\nu - Eg) \tag{1}$$

where $\alpha$ is the absorption coefficient, h is the Planck-constant (h ≈ $4.13567 \times 10^{-15}$ eV·s), $\nu$ is the frequency ($\nu = c/\lambda$, c is velocity of light, c ≈ $3 \times 10^8$ m/s; $\lambda$ is wavelength of the incident light), B is constant, and the exponential n is directly related to the semiconductor type, with $n = 1/2$ for direct bandgap and $n = 2$ for indirect bandgap.

According to Lambert–Beer law, absorbance is proportional to absorption coefficient, that is,

$$A = K\alpha \tag{2}$$

where $A$ is the absorbance of the sample (CuS-I NPs or CuS-I NAs), and K can be regarded as a constant independent of the absorption coefficient.

According to formulas (1) and (2), we can obtain that,

$$(Ah\nu)^{1/n} = BK^{1/n}(h\nu - E_g) \qquad (3)$$

Let $C = BK^{1/n}$, then formula (3) can be rewritten as,

$$(Ah\nu)^{1/n} = C(h\nu - E_g) \qquad (4)$$

If the value of $(Ah\nu)^{1/n}$ is the vertical coordinate and the value of $h\nu$ is the horizontal coordinate, then Eq. (4) can be regarded as a linear equation $y = C(x - E_g)$, and $E_g$ represents the intercept of the line on the $x$ axis in a geometric sense.

## Photothermal effect measurement
The photothermal effect induced by the CuS-I NPs was studied by irradiating various concentrations of CuS-I solution (25, 50, 100, and 200 µg mL$^{-1}$) with NIR laser (808 nm, 2 W cm$^{-2}$, 5 min). The temperature of the solution was measured using an infrared thermal imager. To evaluate the photothermal stability of CuS-I solution, photothermal cycling (five cycles of laser on/off) experiment was carried out. In each cycle, the CuS-I solution (200 µg mL$^{-1}$) was irradiated (808 nm, 5 min), followed by naturally cooling down to room temperature. To evaluate the photothermal effect of CuS-I NAs, CuS-I@P1 (200 µg mL$^{-1}$) treated with furin (20 U mL$^{-1}$) and H$_2$O$_2$ (100 µM) for 8 h at 37 °C, CuS-I@P1 solution (200 µg mL$^{-1}$), and H$_2$O exposed to the 808 nm NIR laser (2 W cm$^{-2}$) for 5 min, respectively. The temperature of the solution was measured using an infrared thermal imager.

## In vitro characterization of furin-H$_2$O$_2$ triggered aggregation of CuS-I@P1 NPs
MDA-MB-468 cells obtained from Procell Life Science&Technology Co.,Ltd (CL-0290B, Wuhan, China) were seeded in culture dishes with a size of 100 mm, and cultured with CuS-I@P1 or CuS-I@P1-scr (200 µg mL$^{-1}$) at 37 °C for 8 h. Then the cells were digested with trypsin and collected, and fixed overnight at 4 °C with 2.5% glutaraldehyde. After elution, embedding, curing, sectioning, and staining, bio-TEM samples were obtained. The pre-treated MDA-MB-468 cells (i.e., co-stained with furin inhibitor II (10 mM) for 1 h) incubated with CuS-I@P1 (200 µg mL$^{-1}$) at 37 °C for 8 h, or 293T cells incubated with CuS-I@P1 (200 µg mL$^{-1}$). Then the cells were digested with trypsin and collected, and fixed overnight at 4 °C with 2.5% glutaraldehyde. After elution, embedding, curing, sectioning, and staining, bio-TEM samples were obtained.

## In vitro time-course confocal fluorescence tests
The pre-treated MDA-MB-468 cells (i.e., co-stained with Golgi Tracker (red) at 37 °C for 0.5 h) incubated with CuS-I@P1 (200 µg mL$^{-1}$) dispersed in serum-free DMEM at 37 °C. And Golgi Tracker was used to reveal the location of Golgi bodies. The obtained Pearson coefficients from the overlay images shown the co-localization of CuS-I@P1 and Golgi body.

## In vitro cytotoxicity tests
MDA-MB-468 cells were cultured on a 96-well plate ($1 \times 10^4$ cells/well). Then, cells were incubated with sequential concentrations of CuS-I@P1 or CuS-I@P1-scr (0, 6.25, 12.5, 25, 50, 100, and 200 µg mL$^{-1}$) for 8 h. Afterward, each cell-containing well was washed with PBS and fresh culture medium was added (100 µL), followed by laser irradiation (808 nm, 2 W cm$^{-2}$, 5 min). Then, cells were cultured for a further 12 h and the cell viability was assessed using MTT. The absorption at 490 nm of the solution in each well was analyzed using a microplate reader after incubation at 37 °C for 4 h.

## Evaluation of intracellular reactive oxygen species of CuS-I@P1
The intracellular ROS generation of CuS-I@P1 was detected using a DCFH-DA kit. MDA-MB-468 cells were incubated with various concentrations of CuS-I@P1 (0, 25, 50, 100 and 200 µg mL$^{-1}$) at 37 °C. Then, cells were washed with PBS three times, DCFH-DA (10 mM, 1 µL) was added into the wells and incubated at 37 °C for 20 min. The cells were washed three times with serum-free cell culture solution to fully remove DCFH-DA that did not enter the cells. The ROS generation capability of CuS-I@P1 was monitored by confocal microscopy. The ROS generation capability of CuS-I@P1 was monitored by confocal microscopy.

## Calcein AM/PI live/dead staining
MDA-MB-468 cells ($2 \times 10^5$ cells/dish) were allowed to adhere to culture dishes and CuS-I@P1 or CuS-I@P1-scr (200 µg mL$^{-1}$) for 8 h. Then, the cells were washed with PBS three times. Afterward, fresh culture medium (800 µL) was added to each dish, and the cells were exposed to laser irradiation (808 nm, 2 W cm$^{-2}$, 5 min). After further incubation for 12 h, the cells were harvested by trypsinization and centrifugation (1000 rpm, 4 min). Next, cells were co-stained with Calcein AM/PI live/dead kit according to the manufacturer's instructions and imaged by confocal microscopy.

## Animal experiments
Animal experiments in this study were strictly conducted according to institutional guidelines and were approved by the Institutional Animal Care and Use Committee of Shanghai Jiao Tong University School of Medicine and Zhejiang University School of Medicine. The institution did not allow mice to have tumors larger than 17 mm in diameter or more than 2000 mm$^3$ in volume. In this experiment, none of the tumor volumes exceed the maximal tumour size/burden. BALB/c-nude mice (female, 6–8 weeks) were purchased from Gempharmatech Co. Ltd (China). All the experimental animals were housed in specific pathogen-free conditions with a 12 h light and 12 h dark cycle, a room temperature of 25 °C and $50.0 \pm 5.0\%$ humidity and had access to food and water ad libitum. When the tumour size reached ~80 mm$^3$, nude mice with MDA-MB-468 tumour xenografts were divided into eight groups randomly ($n = 5$): (1) PBS; (2) PBS + NIR; (3) CuS@P1; (4) CuS@P1 + NIR; (5) CuS-I@P1-scr; (6) CuS-I@P1-scr + NIR; (7) CuS-I@P1; (8) CuS-I@P1 + NIR. PBS (50 µL), CuS@P1 (3 mg mL$^{-1}$, 50 µL), CuS-I@P1-scr (3 mg mL$^{-1}$, 50 µL) and CuS-I@P1 (3 mg mL$^{-1}$, 50 µL) were intratumourally injected every 3 days. After 8 h post-injection, the tumours (groups 2, 4, 6, 8) were exposed to a laser (808 nm, 2 W cm$^{-2}$, 5 min). The body weight and tumour growth were then monitored every 3 days for 36 days. At day 36, tumours were harvested for weighing and sliced for H&E staining. The tumour volume ($V$) was calculated using the formula: $V = L \times W^2/2$, where $W$ is the widest width, and $L$ is the longest length of the tumour.

## Statistics and reproducibility
All data are presented as mean values ± SEM. Unpaired two-tailed Student's $t$ test was used to calculate the P values. Sample size choice was based on previous studies (refs. [4],[8],[10],[16]), not pre-determined by a statistical method. Sample sizes were indicated in the legend of each Figure and Supplementary Fig. No data were excluded. We confirm all attempts at replication were successful. Replicates were conducted for all experiments quantified as described in the Figure legends. Figures [2]a-c, [4]a, [5]a, [5]c, [5]g, [7]e were repeated at least three times and representative example are shown. All samples were randomly allocated into experimental groups. Investigators were not blinded for nanomaterial synthesis, because determination of nanoparticles concentrations are considered as objective measures, not subject to bias. For in vivo experiments, the investigators were blinded to group allocation during data collection and analysis.

## Reporting summary

Further information on research design is available in the Nature Portfolio Reporting Summary linked to this article.

## Data availability

All other data are available from the corresponding authors on request. Source data are provided with this paper.

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

## Acknowledgements

This work is supported by the National Key Research and Development Program of China (2022YFB3203801, 2022YFB3203804, 2022YFB3203800 to D.L. and F.L.), National Natural Science Foundation of China (21936001, 21675001, 21976004, 22376002 to G.W.), National Natural Science Foundation of China (32071374 to F.L.), Anhui Province Outstanding Youth Fund (2008085J10 to G.W.), Leading Talent of "Ten Thousand Plan"-National High-Level Talents Special Support Plan, Program of Shanghai Academic Research Leader under the Science and Technology Innovation Action Plan (21XD1422100 to D.L.), Explorer Program of Science and Technology Commission of Shanghai Municipality (22TS1400700 to D.L.), start-up funds from Shanghai Jiao Tong University (22×010201631 to D.L.), Natural Science Foundation of Zhejiang Province (LR22C100001 to F.L.), Innovative Research Team of High-level Local Universities in Shanghai (SHSMU-ZDCX20210900 to D.L.), CAS Interdisciplinary Innovation Team (JCTD-2020-08 to D.L.), Post-doctoral Innovative Talent Support Program (BX20230220 to Q.W.), Postdoctoral Foundation of China (2023M732244 to Q.W.), Outstanding Innovative Research Team for Molecular Enzymology and Detection in Anhui Provincial Universities (2022AH010012 to G.W.), and Shanghai Municipal Science and Technology Commission (21dz2210100 to D.L.), National Natural Science Foundation of China (22176001 to X.Z.).

## Author contributions

D.L. and G.W. conceived and designed the study. M.X. performed the compound synthesis. M.X., Y.L., G.W., M.G., C.F., Q.W. and X.Z. contributed to nanoparticle characterization and data analyses. B.Z. performed the DFT calculations. M.X., C.F. and C.H. carried out the cell experiments and animal experiments. M.X., S.Y., Y.L. and Q.W. drew the scheme and figures together. D.L., G.W., F.L., M.X., Q.W., Y.L., F.Z., P.L., M.G. and X.Z. co-wrote the manuscript. D.L., G.W., H.L. and F.L. provided project supervision. All the authors approved the final version of the manuscript.

## Competing interests

The authors declare no competing interests.
