## [Peer Review File · Nature Communications]

Reviewers' Comments:

Reviewer #1:

Remarks to the Author:

The authors developed a nanoparticle self-catalysis-regulated in situ assembly approach to achieve dynamic manipulation of nanoparticles in vivo without complicated design and tedious synthesis. The intracellular assembly of CuS-I NPs exhibits activatable dual-modal imaging capability and enhanced photothermal effect, enabling highly efficient imaging and therapy of tumours. Overall, this study is both novel and rigorous, featuring characteristic data and sufficient discussions. Thus, I recommend its publication following minor revisions.

1. The statistical analysis of the whole manuscript is missing, please specify whether these differences are statistically significant.
2. Please specify the function of MeOH and Mannitol mentioned in Figure 1i and j.
3. For the in vitro studies, the authors used H₂O₂ at a concentration of 1 mM, much higher than the known concentration of H₂O₂ in the tumor cells. To achieve tumor-specific furin-instructed intracellular assembly of CuS-I NPs, the concentration of H₂O₂ used for the in vitro studies should be decreased.
4. From Fig. 4c, CuS-I@P1-treated group showed a significantly higher ROS generation at a concentration of 200 µg/mL, which may cause some cell death as observed in Fig. 4g. However, according to Fig. 4d, the cell viability of CuS-I@P1 treated cells was almost 100% even with a concentration up to 200 µg/mL. Please explain this phenomenon.
5. Investigating the photothermal properties of CuS-I NAs relies significantly on understanding their absorption. Therefore, we recommend that the authors present the absorption spectra of CuS-I NAs, and compare it with CuS-I NPs.

Reviewer #2:

Remarks to the Author:

In this manuscript, Xia et al developed a nanoparticle self-catalysis-regulated in situ assembly to achieve dynamic manipulation of nanoparticles in vivo. By introducing Tyr-modified enzyme-responsive peptides as ligands on the surface of I-doped CuS NPs, the controllable, self-propelled assembly of CuS-I NPs via Tyr-Tyr condensation reactions catalyzed by the nanoparticles themselves inside tumour cells was shown. The authors also demonstrated the tumour-specific furin-instructed intracellular formation of CuS-I nano-assemblies in vitro and in vivo experiments, thus exhibiting activatable dual-modal imaging capability and enhanced photothermal therapy.

1 In the manuscript, the author claimed "a nanoparticle self-catalysis-regulated in situ assembly", while the assembled CuS-I NPs are only in the form of aggregation without showing any controlled morphology which may also affect the photothermal role. Thus my big concern is focusing on the assembly mechanism of the nanoparticles via Tyr-Tyr condensation reaction, especially inside the complex environment of tumour cell, and more direct convincing data should be given to support the conclusions, such as the formation of dityrosine bond and the assembled morphology etc.

2 Since the content of Tyr as well as its distribution on the surface of nanoparticles should affect the aggregation, how it affects on both the aggregation size and morphology should be demonstrated in details.

3 The authors claimed that the formation of nanoparticle aggregation could improve the photothermal effect due to "the alignment of CuS-I NAs with photonic-crystal microstructures... and exert substantial influence on the electronic structures of nanomaterials...", again more convincing data should be given.

4 There is a typo on the unit of coordinate of x axis in Figure S5.

Reviewer #3:

Remarks to the Author:

This study presented an intriguing assembly design of CuS-I nanoparticles within tumor cells by

furin protein cleavage and ROS induced dityrosine bonding. The rationale behind this design is based on the overexpression of furin protein in specific tumor cells and Fenton reaction ability of CuS-I nanoparticles when exposed to the elevated levels of H₂O₂ in tumor cells. The cascade reaction, initiated by furin protein cutting ligands and further ROS-triggered tyrosine coupling, leads to the ultimate assembly of CuS-I nanoparticles in tumor cells. The experiments data and characterizations sufficiently proved the successful assembly process. As the same time, a certain properties such as the fluorescence quenching of FAM molecules as they were modified on CuS-I nanoparticles and recovery when liberated by furin protein cutting, are also interesting in this study. Regrettably, besides the enhanced photothermal property of CuS-I nanoparticles, the assembly of CuS-I nanoparticles did not seem to show more significance for tumor therapy through this intricate cascade reaction. On the whole, I think this study is worthy to be published on the Nature Comm. after a minor revise.

Some concerns below:

1. Only from TEM images cannot confirm the S vacancy in CuS-I crystal. The additional characterizations are required for more solid confirmation.
2. In FigS5, the scale on X-axis is absent, as well as the attributions of the vibrations for the important peaks should be labeled in the figure.
3. What is the real level of the concentration in tumor cells? The setting concentration of 1mmol [H₂O₂] in the experiment is too high or not?
4. In Figures12 d, there is an error in the figure legends.
5. In Figure5f, the CuS-I nanocrystals almost did not show obviously photothermal behavior when these nanoparticles were applied on the 293T cells. Even the 293T cells are furin-negative and cannot effectively cleavage the P1 ligands, the separated CuS-I nanocrystals should still have an outstanding photothermal property and can kill the tumor cells under NIR laser irradiation. The authors also presented the photothermal coefficient of CuS-I nanocrystals is as high as 35.6%. So the results in this study is weird to be of no effect for the cytotoxicity of photothermal activation, and it is not consistent to the previous reports of the widely studies of the CuS nanoparticles.
6. In figure 5b, where is the NIR fluorescence signals from?

Responses to the reviewers' comments

Reviewer(s)' Comments to Author:

Reviewer #1

The authors developed a nanoparticle self-catalysis-regulated in situ assembly approach to achieve dynamic manipulation of nanoparticles in vivo without complicated design and tedious synthesis. The intracellular assembly of CuS-I NPs exhibits activatable dual-modal imaging capability and enhanced photothermal effect, enabling highly efficient imaging and therapy of tumours. Overall, this study is both novel and rigorous, featuring characteristic data and sufficient discussions. Thus, I recommend its publication following minor revisions.

Response: *We truly appreciate the reviewer for the encouraging and insightful comments. Based on your suggestions, we have made point-to-point responses and modified the manuscript. We believe that your comments have significantly improved the quality of our manuscript.*

1. The statistical analysis of the whole manuscript is missing, please specify whether these differences are statistically significant.

Response: *Thank you for your kind comment. We have performed the statistical analysis on Figures 4d-f, Figures 5c, e and Figure 6d. The statistical significance and analysis methods are presented in the updated figures and figure legends.*

Our modification to the manuscript: *The statistical analysis was performed on Figures 4d-f, Figures 5c, e and Figure 6d.*

- Figure 4d-f

Figure 4. d,e, Cell viability of MDA-MB-468 cells incubated with different concentrations (0-200 $\mu\text{g mL}^{-1}$) of CuS-I@P1 (**d**) or CuS-I@P1-scr (**e**) for 8 h with (or without) NIR irradiation. **f,** Cell viability of 293T cells incubated with different concentrations (0-200 $\mu\text{g mL}^{-1}$) of CuS-I@P1 for 8 h with (or without) NIR irradiation. All data are presented as means \pm SEM, $n = 3$ independent experiments. Statistical significance was analyzed by unpaired, two-tailed Student's t-tests (**d-f**). * $P < 0.05$; n.s., no significant difference.

- Figure 5c and 5e

Figure 5. **c**, The time-dependent quantitative calculation of the average fluorescence intensity at the tumour area. **e**, The average PA intensity increment at 830 nm ($\Delta PA_{\text{average}}$) of tumour areas after intratumour administration of CuS@P1, CuS-I@P1 and CuS-I@P1-scr. All data are presented as means \pm SEM, $n=3$ independent experiments. Statistical significance was analyzed by unpaired, two-tailed Student's t-tests. **** $P < 0.0001$, *** $P < 0.001$, ** $P < 0.01$.

- Figure 6d

Figure 6. **d**, Relative tumour volume curves of MDA-MB-468 tumour-bearing mice under distinct treatments for 36 d. All data are presented as means \pm SEM, $n=5$ independent experiments. Statistical significance was analyzed by unpaired, two-tailed Student's t-tests. * $P < 0.05$.

2. Please specify the function of MeOH and Mannitol mentioned in Figure 1i and j.

Response: Thank you for your valuable comment. In Figure 1e, we have confirmed the generation of hydroxyl radicals ($\bullet OH$) by CuS-I NPs, which serve as catalysts for tyrosine dimerization. To further substantiate the pivotal role of $\bullet OH$ in the tyrosine dimerization process, we employed methanol (MeOH) and mannitol as $\bullet OH$ scavengers for their removal. Our results demonstrate that the $\bullet OH$ generated during the reaction of CuS-I NPs with H_2O_2 plays a crucial role in tyrosine dimerization.

Our modification to the manuscript: The following sentence was modified on page 5 of the revised manuscript.

- Page 5

“Addition of reactive oxygen quenching agents (methanol or mannitol) significantly inhibits the formation of dityrosine (Fig. 1i, j), suggesting the indispensability of $\bullet OH$ generated by CuS-I NPs in Tyr dimerization.”

3. For the in vitro studies, the authors used H_2O_2 at a concentration of 1 mM, much higher than the known concentration of H_2O_2 in the tumor cells. To achieve tumor-specific furin-instructed

intracellular assembly of CuS-I NPs, the concentration of H_2O_2 used for the in vitro studies should be decreased.

Response: Thank you very much for your suggestion. According to your comment, we selected $100 \mu M H_2O_2$, similar to the levels found in tumour cells (*Adv. Sci.*, 2023, 10, 2301919), as the source of ROS for the in vitro studies. As illustrated in the revised figures, CuS-I NPs maintain their remarkable catalytic performance in facilitating the dimerization of tyrosine, even when the H_2O_2 concentration is reduced to $100 \mu M$.

Our modification to the manuscript: The in vitro studies performed in $100 \mu M H_2O_2$ were added as Figures 1f, h-k, Figures 3a-c, h, i and Supplementary Fig. 10. in the revised manuscript. The following sentences were modified on pages 5, 7, 8, 14 and 16 of the revised manuscript.

- Figure 1

Figure 1. Characterization and catalytic performance of CuS-I NPs. **a**, Transmission electron

microscopy (TEM) image of CuS-I NPs with the corresponding SAED pattern shown as the inset. **b**, EDX elemental mapping of CuS-I NPs. **c**, HRTEM image of CuS-I NPs, with vacancy defects marked by the white circles. **d**, XRD spectrum of CuS NPs and CuS-I NPs. **e**, EPR spectrum of 5, 5-dimethyl-1-pyrroline-N-oxide (DMPO)-•OH spin adducts generated by CuS-I NPs and CuS NPs in the presence of H₂O₂ using DMPO as the •OH trapping agent. **f**, UV-Vis absorbance spectrum of MB incubated with CuS NPs and CuS-I NPs in the presence of H₂O₂. **g**, Mechanism of the Tyr dimerization catalyzed by CuS-I NPs under a mild condition mimicking tumour intracellular microenvironment (phosphate buffer solution (PBS) buffer containing 100 μM H₂O₂, and pH = 6.5). **h**, EPR spectrum of DMPO-tyrosyl radical spin adducts when Tyr treated with CuS-I NPs in the presence of H₂O₂. **i,j**, Fluorescence emission curves (**i**) and UV-Vis absorbance spectrum (**j**) of Tyr + H₂O₂ after different treatments. **k**, HPLC trace of Tyr and Tyr treated with CuS-I NPs in the presence of H₂O₂.

- Page 5

“In the presence of H₂O₂, the absorbance variation of methylene blue (MB) at 664 nm³⁷ mixed with CuS-I NPs is 2.03-fold lower than that of CuS NPs, demonstrating that I doping significantly enhances the Fenton-like reaction to produce •OH (Fig. 1f).”

• Figure 3

Figure 3. Design and characterization of self-catalysis-instructed self-propelled assembly of CuS-I@P1 NPs. **a**, Scheme of the self-propelled in situ assembly of CuS-I NPs into CuS-I NAs and TEM images of CuS-I@P1 NPs before (left) and after (right) treated with furin and H₂O₂ for 8 h. **b**, Hydrodynamic size distribution of CuS-I@P1 NPs incubated with furin and H₂O₂ for 0, 2, 4, 6, and 8 h. **c**, Fluorescence signals of dityrosine ($\lambda_{ex} = 315$ nm) after CuS-I@P1, CuS@P1 and CuS-I@P1-scr incubated with furin-H₂O₂ for different times. **d**, Fluorescence intensities of dityrosine ($\lambda_{ex} = 315$ nm) after CuS-I@P1 incubated with metalloprotease, Cathepsin B, Caspase-3 (Casp-3), alkaline phosphatase (ALP) and furin in the presence of H₂O₂. **e**, Fluorescence emission spectrum of FAM under the same peptide concentration (3 $\mu\text{g mL}^{-1}$) ($\lambda_{ex} = 490$ nm). **f**, Fluorescence emission spectrum of FAM in CuS-I@P1 NPs solution before and after treatment with furin. **g**, Fluorescence emission spectrum of FAM in CuS-I@P1 NPs solution with different peptide concentrations after treatment with furin. **h**, Infrared thermal images of H₂O, CuS-I NPs, and CuS-I NAs solution exposed to laser irradiation for 5 min (808 nm, 2 W cm⁻²). **i**, Temperature change curves of H₂O, CuS-I NPs and CuS-I NAs

solution with laser irradiation for 5 min (808 nm, 2 W cm⁻²). All data are presented as means ± SEM, n = 3 independent experiments. Statistical significance was analyzed by unpaired, two-tailed Student's t-tests. ****P < 0.0001.

- Supplementary Fig. 10

Supplementary Fig.10. TEM images of CuS-I@P1 incubated with furin and H₂O₂ for different times ([CuS-I@P1] = 200 μg mL⁻¹, [H₂O₂] = 100 μM, [furin] = 20 U mL⁻¹).

- Page 7

“After treatment with furin and H₂O₂, the average hydrodynamic diameter of CuS-I@P1 increases from ~14.17 nm to ~79.85 nm (Fig. 3b, Supplementary Fig. 10),”

- Page 8

“Upon exposure to an 808 nm laser at 2 W cm⁻² for 5 minutes, the temperature of CuS-I NAs increases by 29.2 °C, exhibiting a close to two-fold temperature elevation compared to CuS-I@P1 NPs (Fig. 3h, i).”

- Page 14

“Firstly, CuS-I NPs or CuS NPs solution (1 mg mL⁻¹, 200 μL) was added into MB solution (5 mg mL⁻¹, 20 μL) in the presence of H₂O₂ (100 μM, 780 μL)”

- Page 14

“In the presence of H₂O₂ (100 μM, 1 mL), the Tyr solution (1 mg mL⁻¹, 200 μL) was mixed with CuS (1 mg mL⁻¹, 800 μL),”

- Page 14

“In the presence of H₂O₂ (100 μM), CuS-I@P1, CuS-I@P1-scr, CuS@P1 (200 μg mL⁻¹) were mixed with furin (20 U mL⁻¹), then the fluorescence intensity of the mixed solution was measured at different times by fluorescence spectrophotometer (λ_{ex} = 315 nm). In the presence of H₂O₂ (100 μM), CuS-I@P1 solution (200 μg mL⁻¹) were mixed with furin, metalloprotease, Cathepsin B, Caspase 3, ALP for 8 h, then the fluorescence intensity of the mixed solution was measured by a fluorescence spectrophotometer (λ_{ex} = 315 nm).”

“HPLC traces of Tyr (0.1 mg mL⁻¹), Tyr (1 mg mL⁻¹, 200 μL) treated with CuS-I solution (1 mg mL⁻¹, 800 μL) and H₂O₂ (100 μM, 1 mL) for 8 h at 37 °C were detected.”

- Page 16

“To evaluate the photothermal effect of CuS-I NAs, CuS-I@P1 (200 μg mL⁻¹) treated with furin (20 U mL⁻¹) and H₂O₂ (100 μM) for 8 h at 37 °C”

4. From Fig. 4c, CuS-I@P1-treated group showed a significantly higher ROS generation at a concentration of 200 μg/mL, which may cause some cell death as observed in Fig. 4g. However,

according to Fig. 4d, the cell viability of CuS-I@P1 treated cells was almost 100% even with a concentration up to 200 $\mu\text{g}/\text{mL}$. Please explain this phenomenon.

Response: Thank you for your insightful comment. We apologize for any errors resulting from past experimental procedures, which may be attributed to operational deficiencies during the removal of the culture medium with MTT, leaving Formazan. We cautiously repeated the experiment, revealing a gradual decrease in cell viability with an increasing dosage of CuS-I@P1 NPs in the absence of laser irradiation. Cell viability remained at $\sim 70\%$ with a dosage of 200 $\mu\text{g mL}^{-1}$.

Our modification to the manuscript: The updated Figure 4d was added on page 32 of the revised manuscript.

- Figure 4d

Figure 4. Cell viability of MDA-MB-468 cells incubated with different concentrations (0-200 $\mu\text{g mL}^{-1}$) of CuS-I@P1 (d) for 8 h with (or without) NIR irradiation. All data are presented as means \pm SEM, $n=3$ independent experiments. Statistical significance was analyzed by unpaired, two-tailed Student's t-tests (d). * $P < 0.05$.

5. Investigating the photothermal properties of CuS-I NAs relies significantly on understanding their absorption. Therefore, we recommend that the authors present the absorption spectra of CuS-I NAs, and compare it with CuS-I NPs.

Response: Thank you for your kind comment. Based on your helpful suggestion, we conducted absorption spectrum measurements for CuS-I NAs and CuS-I NPs. Notably, CuS-I NAs exhibit enhanced light absorption capacity at 808 nm compared to CuS-I NP. This enhancement promotes carrier transition during laser excitation (Langmuir, 2014, 30, 1416-1423; Dalton Trans., 2019, 48, 4495-4503), thus improving the photothermal effect.

Our modification to the manuscript: The absorption spectra of CuS-I NPs and CuS-I NAs were added as Supplementary Fig. 15 in the revised manuscript. The following sentence was added on page 8 of the revised manuscript.

- Page 8

“Moreover, the self-propelled assembly of CuS-I NPs can further improve the photothermal effect. The light absorption of CuS-I NAs at 808 nm was enhanced compared to that of CuS-I NPs (Supplementary Fig. 15).”

- Supplementary Fig. 15

Supplementary Fig.15. The absorption spectra of CuS-I NPs and CuS-I NAs.

Thank you very much for the valuable comments, which have significantly improved the quality of the manuscript.

Reviewer #2

In this manuscript, Xia et al developed a nanoparticle self-catalysis-regulated in situ assembly to achieve dynamic manipulation of nanoparticles in vivo. By introducing Tyr-modified enzyme-responsive peptides as ligands on the surface of I-doped CuS NPs, the controllable, self-propelled assembly of CuS-I NPs via Tyr-Tyr condensation reactions catalyzed by the nanoparticles themselves inside tumour cells was shown. The authors also demonstrated the tumour-specific furin-instructed intracellular formation of CuS-I nano-assemblies in vitro and in vivo experiments, thus exhibiting activatable dual-modal imaging capability and enhanced photothermal therapy.

Response: *We truly appreciate the reviewer for the encouraging and insightful comments. Based on your suggestions, we have made point-to-point responses and modified the manuscript. We believe that your comments have significantly improved the quality of our manuscript.*

1. In the manuscript, the author claimed “a nanoparticle self-catalysis-regulated in situ assembly”, while the assembled CuS-I NPs are only in the form of aggregation without showing any controlled morphology which may also affect the photothermal role. Thus my big concern is focusing on the assembly mechanism of the nanoparticles via Tyr-Tyr condensation reaction, especially inside the complex environment of tumour cell, and more direct convincing data should be given to support the conclusions, such as the formation of dityrosine bond and the assembled morphology etc.

Response: *Thank you for your constructive comment. Following your suggestion, we characterized the assembled morphology of CuS-I@PI NPs and the formation of dityrosine bonds inside the complex environment of tumour cells. CuS-I@PI NPs were incubated with MDA-MB-468 cells for 8 hours. Subsequently, the cells were trypsinized, and the cell membrane was disrupted using via a freeze-thaw process to prepare cell lysate (Toxins, 2021, 13, 596), which ensures that temperature changes do not disrupt the covalent cross-linking-based CuS-I NAs. TEM observation revealed the assembly of CuS-I@PI NPs in the cell lysate,*

and typical fluorescence emission associated with dityrosine was detected. These results demonstrate that the assembly of CuS-I@P1 NPs in tumour cells is achieved through the self-catalysis-regulated Tyr dimerization.

Our modification to the manuscript: The TEM image of CuS-I@P1 NPs incubated with MDA-MB-468 cells and the fluorescence intensity of dityrosine in the MDA-MB-468 cell lysate were added to Supplementary Fig. 18 in the revised supporting information. The following sentences were added on page 9 of the revised manuscript.

- Page 9

“TEM observation revealed the assembly of CuS-I@P1 NPs in the lysate of MDA-MB-468 cells, along with the typical fluorescence emission at ~ 410 nm associated with dityrosine (Supplementary Fig. 18). Also, we observed large areas of assemblies within cells treated with CuS-I@P1 NPs (Supplementary Fig. 19a), while only a few CuS-I@P1-scr NPs remained in the cytoplasm of CuS-I@P1-scr-treated cells (Supplementary Fig. 19b). These results further verify that the assembly of CuS-I@P1 NPs in tumour cells is achieved through self-catalysis-regulated Tyr dimerization.”

- Supplementary Fig. 18

Supplementary Fig.18. (a) TEM image of the MDA-MB-468 cell lysate after incubation with CuS-I@P1 NPs for 8 h at 37 °C. (b) The fluorescence spectrum of the lysate of MDA-MB-468 cells, following an 8-hour incubation with CuS-I@P1 NPs at 37 °C, exhibits a characteristic fluorescence signal of dityrosine at ~ 410 nm.

2 Since the content of Tyr as well as its distribution on the surface of nanoparticles should affect the aggregation, how it affects on both the aggregation size and morphology should be demonstrated in details.

Response: Thank you for your kind comment. Indeed, the covalent cross-linking-based nanoparticle assembly is closely related to surface ligands, whose size and morphology typically increase and reach a plateau as the feeding amount of surface ligands increases (*Adv. Funct. Mater.*, 2020, 30, 2001566). Accordingly, we modified the surface of CuS-I NPs by introducing varying quantities of Tyr and monitored the resulting changes in aggregation size and morphology. CuS-I NPs with distinct Tyr contents were prepared by adding varying volumes of a 0.55 mM Tyr solution (10 μ L, 20 μ L, 40 μ L, 60 μ L, 80 μ L, 100 μ L), denoted as CuS-I@Tyr-1, CuS-I@Tyr-2, CuS-I@Tyr-3, CuS-I@Tyr-4, CuS-I@Tyr-5, and CuS-I@Tyr-6, respectively. Upon treatment with H₂O₂ (100 μ M), the morphology and size of the nanoparticles were examined using TEM. TEM images revealed a gradual increase in

assembly size as the feeding amount of Tyr increased, ultimately reaching a plateau, which is consistent with the trend reported in previous article.

Our modification to the manuscript: TEM images of CuS-I NPs with distinct Tyr contents upon treatment with H₂O₂ were added to Supplementary Fig. 9 in the revised supporting information. The following sentences were added on page 7 of the revised manuscript.

- Page 7

“Furthermore, we optimized the coupling of Tyr on CuS-I NPs by introducing varying quantities of Tyr. It was observed that as the Tyr feeding amount increased, the assembly size of CuS-I NPs exhibited a gradual initial increase followed by stabilization (Supplementary Fig. 9).”

- Supplementary Fig. 9

Supplementary Fig.9. TEM images of CuS-I NPs with distinct Tyr contents upon treatment with H₂O₂, which were prepared by adding varying volumes of a 0.55 mM Tyr solution (10 μ L, 20 μ L, 40 μ L, 60 μ L, 80 μ L, 100 μ L), denoted as CuS-I@Tyr-1, CuS-I@Tyr-2, CuS-I@Tyr-3, CuS-I@Tyr-4, CuS-I@Tyr-5, and CuS-I@Tyr-6, respectively.

3 The authors claimed that the formation of nanoparticle aggregation could improve the photothermal effect due to “the alignment of CuS-I NAs with photonic-crystal microstructures... and exert substantial influence on the electronic structures of

nanomaterials...”, again more convincing data should be given.

Response: Thank you for your insightful comment. The molar extinction coefficient (ϵ), closely linked to electronic structures, is the primary factor contributing to the variation in temperature increments. In the context of semiconductor materials, a relationship can be established between the optical band gap (E_g) and the absorption coefficient ($\alpha = \epsilon/M$), expressed as,

$$(\alpha h\nu)^{1/n} = B(h\nu - E_g) \quad (1)$$

where h is the Planck-constant ($h \approx 4.13567 \times 10^{-15} \text{ eV}\cdot\text{s}$), ν is the frequency ($\nu = c/\lambda$, c is velocity of light, $c \approx 3 \times 10^8 \text{ m/s}$; λ is wavelength of the incident light), B is constant, and the exponential n is directly related to the semiconductor type, with $n = 1/2$ for direct band gap and $n = 2$ for indirect band gap. Accordingly, we calculated the optical band gap of CuS NPs and CuS-I NAs to illustrate the enhanced photothermal effect mechanisms exhibited by CuS-I NAs.

The absorption of CuS NPs stems from the $d-d$ transition of Cu^{2+} ions, resulting in both a direct band gap at $\sim 500 \text{ nm}$ and an indirect one in the NIR region (*J. Am. Chem. Soc.*, 2009, 131, 4253-4261). The distance-dependent interaction among the electronic structures of CuS NPs significantly impacts the indirect transition and, consequently, the absorption intensity of CuS NPs at 808 nm (*Nanomedicine*, 2010, 5, 1161-1171).

According to Lambert-Beer law, absorbance is proportional to absorption coefficient, that is

$$A = K\alpha \quad (2)$$

where A is the absorbance of the sample (CuS-I NPs or CuS-I NAs), and K can be regarded as a constant independent of the absorption coefficient.

According to formulas (1) and (2), we can obtain that,

$$(A h\nu)^{1/n} = BK^{1/n}(h\nu - E_g) \quad (3)$$

Let $C = BK^{1/n}$, then formula (3) can be rewritten as,

$$(A h\nu)^{1/n} = C(h\nu - E_g) \quad (4)$$

If the value of $(A h\nu)^{1/n}$ is the vertical coordinate and the value of $h\nu$ is the horizontal coordinate, then equation (4) can be regarded as a linear equation $y = C(x - E_g)$, and E_g represents the intercept of the line on the X -axis in a geometric sense.

We substituted the measured absorbance values of CuS-I NPs and CuS-I NAs (Supplementary Fig. 15) into the above formulas for conversion and calculation. The calculation revealed a bandgap of 2.03 eV for CuS-I NPs, aligning with a prior study (*J. Alloys Compd.*, 2021, 864, 158591). Additionally, the bandgap of CuS-I NAs is 1.81 eV , lower than that of dispersed CuS-I NPs. These findings suggest a reduced bandgap in CuS-I NAs, facilitating $d-d$ transitions of Cu^{2+} ions and enhancing the absorption in the NIR region. This characteristic is beneficial for augmenting the photothermal effect.

Our modification to the manuscript: The band gaps of CuS-I NPs and CuS-I NAs were added to Supplementary Fig. 16 in the supporting information. The following sentences were added on pages 8 and 15 of the revised manuscript.

- Page 8

“As shown in Supplementary Fig. 16, CuS-I NAs exhibited a bandgap of 1.81 eV , which is lower than the 2.03 eV observed for CuS-I NPs, resulting in an augmented molar extinction

coefficient and improved light absorption at 808 nm in comparison to dispersed CuS-I NPs⁵².”

Reference

52 Wang, R., Shan, G., Wang, T., Yin, D. & Chen, Y. Photothermal enhanced photocatalytic activity based on Ag-doped CuS nanocomposites. *J. Alloys Compd.* **864**, 158591, (2021).

- Supplementary Fig. 16

Supplementary Fig.16. Band gaps (E_g) for CuS-I NPs and CuS-I NAs were determined using the Tauc plot method.

- Page 15

“**Calculation of band gaps for CuS-I NPs and CuS-I NAs.** We utilized the Tauc plot method to calculate the band gaps (E_g) of CuS-I NPs and CuS-I NAs.

$$\text{Tauc plot: } (\alpha h\nu)^{1/n} = B(h\nu - E_g) \quad (1)$$

where α is the absorption coefficient, h is the Planck-constant ($h \approx 4.13567 \times 10^{-15} \text{ eV}\cdot\text{s}$), ν is the frequency ($\nu = c/\lambda$, c is velocity of light, $c \approx 3 \times 10^8 \text{ m/s}$; λ is wavelength of the incident light), B is constant, and the exponential n is directly related to the semiconductor type, with $n = 1/2$ for direct band gap and $n = 2$ for indirect band gap.

According to Lambert-Beer law, absorbance is proportional to absorption coefficient, that is

$$A = K\alpha \quad (2)$$

where A is the absorbance of the sample (CuS-I NPs or CuS-I NAs), and K can be regarded as a constant independent of the absorption coefficient.

According to formulas (1) and (2), we can obtain that,

$$(Ah\nu)^{1/n} = BK^{1/n}(h\nu - E_g) \quad (3)$$

Let $C = BK^{1/n}$, then formula (3) can be rewritten as,

$$(Ah\nu)^{1/n} = C(h\nu - E_g) \quad (4)$$

If the value of $(Ah\nu)^{1/n}$ is the vertical coordinate and the value of $h\nu$ is the horizontal coordinate, then equation (4) can be regarded as a linear equation $y = C(x - E_g)$, and E_g represents the intercept of the line on the X-axis in a geometric sense.”

4 There is a typo on the unit of coordinate of x axis in Figure S5.

Response: Thank you for your careful attention to detail. We apologize for the error on the unit of coordinate of x axis in the previous Supplementary Fig. 5 (modified Supplementary Fig. 6) and have revised the mistake.

Our modification to the manuscript: The unit of coordinate of x axis in Supplementary Fig. 6a was modified in the revised supporting information.

- Supplementary Fig. 6

Supplementary Fig.6. (a) Fourier infrared (FT-IR) spectrum of CuS-I NPs, P1 and CuS-I@P1 NPs.

Thank you very much for the valuable comments, which have significantly improved the quality of the manuscript.

Reviewer #3

This study presented an intriguing assembly design of CuS-I nanoparticles within tumor cells by furin protein cleavage and ROS induced dityrosine bonding. The rationale behind this design is based on the overexpression of furin protein in specific tumor cells and Fenton reaction ability of CuS-I nanoparticles when exposed to the elevated levels of H₂O₂ in tumor cells. The cascade reaction, initiated by furin protein cutting ligands and further ROS-triggered tyrosine coupling, leads to the ultimate assembly of CuS-I nanoparticles in tumor cells. The experiments data and characterizations sufficiently proved the successful assembly process. As the same time, certain properties such as the fluorescence quenching of FAM molecules as they were modified on CuS-I nanoparticles and recovery when liberated by furin protein cutting, are also interesting in this study. Regrettably, besides the enhanced photothermal property of CuS-I nanoparticles, the assembly of CuS-I nanoparticles did not seem to show more significance for tumor therapy through this intricate cascade reaction. On the whole, I think this study is worthy to be published on the Nature Comm. after a minor revise.

Response: *We sincerely appreciate the reviewer for the encouraging and insightful comments. Based on your suggestions, we have made point-to-point responses and modified the manuscript. We believe that your comments have significantly improved the quality of our manuscript.*

1. Only from TEM images cannot confirm the S vacancy in CuS-I crystal. The additional characterizations are required for more solid confirmation.

Response: *Thank you for your insightful comment. To prove the existence of sulfur (S) vacancies more directly and comprehensively, we used electron paramagnetic resonance (EPR) to further characterize the structure of CuS-I crystal, which revealed an obvious EPR signal belonging to S vacancies in the CuS-I crystal.*

Our modification to the manuscript: *The EPR spectra were added as Supplementary Fig. 4 in the revised supporting information.*

- Page 4

“Compared with CuS NPs (Supplementary Fig. 3), CuS-I NPs show a higher density of S defects, which can enhance their catalytic performance by increasing the active sites on the surface (Fig. 1c)^{34,35}. This observation aligns with the electron paramagnetic resonance (EPR) pattern, which reveals more pronounced S vacancies in CuS-I NPs³⁶ (Supplementary Fig. 4).”

Reference

36 Hu, J. et al. Sulfur vacancy-rich MoS₂ as a catalyst for the hydrogenation of CO₂ to methanol. *Nat. Catal.* **4**, 242-250, (2021).

- Supplementary Fig. 4

Supplementary Fig.4. Electron paramagnetic resonance (EPR) spectra of CuS NPs and CuS-I NPs.

2. In FigS5, the scale on X-axis is absent, as well as the attributions of the vibrations for the important peaks should be labeled in the figure.

Response: Thank you for your careful attention to detail. We apologize for the error on the x axis in the previous Supplementary Fig. 5 (modified Supplementary Fig. 6) and have revised the mistake. Moreover, the attributions of the vibrations for the important peaks were labeled in the modified figure.

Our modification to the manuscript: The Supplementary Fig. 6a was modified in the revised supporting information.

- Supplementary Fig. 6

Supplementary Fig.6. (a) Fourier infrared (FT-IR) spectrum of CuS-I NPs, P1 and CuS-I@P1 NPs.

3. What is the real level of the concentration in tumor cells? The setting concentration of 1mmol [H₂O₂] in the experiment is too high or not?

Response: Thank you for your insightful comments. The H₂O₂ concentration in tumour cells

ranges from 100×10^{-6} to 1×10^{-3} M (*Adv. Sci.*, 2023, 10, 2301919), indicating that selecting 1mM H_2O_2 to mimic the internal environment of tumour cells in the *in vitro* experiments is relatively high. Therefore, we selected 100 μ M H_2O_2 , similar to the levels found in tumour cells, as the source of ROS for the *in vitro* studies. As illustrated in the revised figures, CuS-I NPs maintain their remarkable catalytic performance in facilitating the dimerization of tyrosine, even when the H_2O_2 concentration is reduced to 100 μ M.

Our modification to the manuscript: The *in vitro* studies performed in 100 μ M H_2O_2 were added as Figures. 1f, h-k, Figures. 3a-c, h, i and Supplementary Fig. 10. in the revised manuscript. The following sentences were modified on pages 5, 7, 8, 14 and 16 of the revised manuscript.

- Figure 1

Figure 1. Characterization and catalytic performance of CuS-I NPs. **a**, Transmission electron microscopy (TEM) image of CuS-I NPs with the corresponding SAED pattern shown as the

inset. **b**, EDX elemental mapping of CuS-I NPs. **c**, HRTEM image of CuS-I NPs, with vacancy defects marked by the white circles. **d**, XRD spectrum of CuS NPs and CuS-I NPs. **e**, EPR spectrum of 5, 5-dimethyl-1-pyrroline-N-oxide (DMPO)-•OH spin adducts generated by CuS-I NPs and CuS NPs in the presence of H₂O₂ using DMPO as the •OH trapping agent. **f**, UV-Vis absorbance spectrum of MB incubated with CuS NPs and CuS-I NPs in the presence of H₂O₂. **g**, Mechanism of the Tyr dimerization catalyzed by CuS-I NPs under a mild condition mimicking tumour intracellular microenvironment (phosphate buffer solution (PBS) buffer containing 100 μM H₂O₂, and pH = 6.5). **h**, EPR spectrum of DMPO-tyrosyl radical spin adducts when Tyr treated with CuS-I NPs in the presence of H₂O₂. **i,j**, Fluorescence emission curves (**i**) and UV-Vis absorbance spectrum (**j**) of Tyr + H₂O₂ after different treatments. **k**, HPLC trace of Tyr and Tyr treated with CuS-I NPs in the presence of H₂O₂.

- Page 5

“In the presence of H₂O₂, the absorbance variation of methylene blue (MB) at 664 nm³⁷ mixed with CuS-I NPs is 2.03-fold lower than that of CuS NPs, demonstrating that I doping significantly enhances the Fenton-like reaction to produce •OH (Fig. 1f).”

• Figure 3

Figure 3. Design and characterization of self-catalysis-instructed self-propelled assembly of CuS-I@P1 NPs. **a**, Scheme of the self-propelled in situ assembly of CuS-I NPs into CuS-I NAs and TEM images of CuS-I@P1 NPs before (left) and after (right) treated with furin and H₂O₂ for 8 h. **b**, Hydrodynamic size distribution of CuS-I@P1 NPs incubated with furin and H₂O₂ for 0, 2, 4, 6, and 8 h. **c**, Fluorescence signals of dityrosine (λ_{ex} = 315 nm) after CuS-I@P1, CuS@P1 and CuS-I@P1-scr incubated with furin-H₂O₂ for different times. **d**, Fluorescence intensities of dityrosine (λ_{ex} = 315 nm) after CuS-I@P1 incubated with metalloprotease, Cathepsin B, Caspase-3 (Casp-3), alkaline phosphatase (ALP) and furin in the presence of H₂O₂. **e**, Fluorescence emission spectrum of FAM under the same peptide concentration (3 μg mL⁻¹) (λ_{ex} = 490 nm). **f**, Fluorescence emission spectrum of FAM in CuS-I@P1 NPs solution before and after treatment with furin. **g**, Fluorescence emission spectrum of FAM in CuS-I@P1 NPs solution with different peptide concentrations after treatment with furin. **h**, Infrared

thermal images of H₂O, CuS-I NPs, and CuS-I NAs solution exposed to laser irradiation for 5 min (808 nm, 2 W cm⁻²). **i**, Temperature change curves of H₂O, CuS-I NPs and CuS-I NAs solution with laser irradiation for 5 min (808 nm, 2 W cm⁻²). All data are presented as means ± SEM, n = 3 independent experiments. Statistical significance was analyzed by unpaired, two-tailed Student's t-tests. **** $P < 0.0001$.

- Supplementary Fig. 10

Supplementary Fig.10. TEM images of CuS-I@P1 incubated with furin and H₂O₂ for different times ([CuS-I@P1] = 200 μg mL⁻¹, [H₂O₂] = 100 μM, [furin] = 20 U mL⁻¹).

- Page 7

“After treatment with furin and H₂O₂, the average hydrodynamic diameter of CuS-I@P1 increases from ~14.17 nm to ~79.85 nm (Fig. 3b, Supplementary Fig. 10),”

- Page 8

“Upon exposure to an 808 nm laser at 2 W cm⁻² for 5 minutes, the temperature of CuS-I NAs increases by 29.2 °C, exhibiting a close to two-fold temperature elevation compared to CuS-I@P1 NPs (Fig. 3h, i).”

- Page 14

“Firstly, CuS-I NPs or CuS NPs solution (1 mg mL⁻¹, 200 μL) was added into MB solution (5 mg L⁻¹, 20 μL) in the presence of H₂O₂ (100 μM, 780 μL)”

- Page 14

“In the presence of H₂O₂ (100 μM, 1 mL), the Tyr solution (1 mg mL⁻¹, 200 μL) was mixed with CuS (1 mg mL⁻¹, 800 μL)”

- Page 14

“In the presence of H₂O₂ (100 μM), CuS-I@P1, CuS-I@P1-scr, CuS@P1 (200 μg mL⁻¹) were mixed with furin (20 U mL⁻¹), then the fluorescence intensity of the mixed solution was measured at different times by fluorescence spectrophotometer (λ_{ex} = 315 nm). In the presence of H₂O₂ (100 μM), CuS-I@P1 solution (200 μg mL⁻¹) were mixed with furin, metalloprotease, Cathepsin B, Caspase 3, ALP for 8 h, then the fluorescence intensity of the mixed solution was measured by a fluorescence spectrophotometer (λ_{ex} = 315 nm).”

“HPLC traces of Tyr (0.1 mg mL⁻¹), Tyr (1 mg mL⁻¹, 200 μL) treated with CuS-I solution (1 mg mL⁻¹, 800 μL) and H₂O₂ (100 μM, 1 mL) for 8 h at 37 °C were detected.”

- Page 16

“To evaluate the photothermal effect of CuS-I NAs, CuS-I@P1 (200 μg mL⁻¹) treated with furin (20 U mL⁻¹) and H₂O₂ (100 μM) for 8 h at 37 °C”

4. In Figures S12d, there is an error in the figure legends.

Response: Thank you for your kind comment. We apologize for the error in the figure legends of the previous Supplementary Fig. 12d (modified Supplementary Fig. 14d) and have revised the mistake.

Our modification to the manuscript: The legends in Supplementary Fig. 14d were modified in the revised supporting information.

- Supplementary Fig. 14d

Supplementary Fig.14. (d) Temperature change curves of CuS-I solution at different power densities ([CuS-I] = 200 $\mu\text{g mL}^{-1}$).

5. In Figure 5f, the CuS-I nanocrystals almost did not show obviously photothermal behavior when these nanoparticles were applied on the 293T cells. Even the 293T cells are furin-negative and cannot effectively cleavage the P1 ligands, the separated CuS-I nanocrystals should still have an outstanding photothermal property and can kill the tumor cells under NIR laser irradiation. The authors also presented the photothermal efficiency of CuS-I nanocrystals is as high as 35.6%. So the results in this study is weird to be of no effect for the cytotoxicity of photothermal activation, and it is not consistent to the previous reports of the widely studies of the CuS nanoparticles.

Response: Thank you for your kind comment. We repeated the experiment and performed live/dead staining, revealing the high safety of CuS-I NPs for 293T cells. It is known that the modification of NPs with a peptide containing the RVR sequence, recognized for targeting tumour cells with elevated furin expression, leads to insufficient uptake by cells with low furin expression (Nat. Mater., 2019, 18, 1376-1383; Anal. Chem. 2021, 93, 9277-9285). Consequently, the limited uptake by furin-negative 293T cells may be the reason for the high safety of CuS-I NPs on these cells.

Our modification to the manuscript: The Figure 4f and Supplementary Fig. 20. were repeated, and the updated data were added in the revised manuscript.

- Figure 4f

Figure 4. f. Cell viability of 293T cells incubated with different concentrations (0-200 µg mL⁻¹) of CuS-I@P1 for 8 h with (or without) NIR irradiation.

- Supplementary Fig. 20

Supplementary Fig.20. Confocal fluorescence images of 293T cells co-stained with calcein AM (green, live cells) and propidium iodide (red, dead cells) after treated with phosphate buffer solution (PBS), CuS-I@P1, CuS-I@P1-scr with or without NIR laser irradiation ([CuS-I@P1] = [CuS-I@P1-scr] = 200 µg mL⁻¹).

6. In figure 5b, where is the NIR fluorescence signals from?

Response: Thank you for your kind comment. We apologize for the imprecise description we made. The accurate expression should delete “NIR”, because the fluorescence signals originate from 5-Carboxyfluorescein of response short peptides (FAM-KRVRRY) with the emission peak at ~520 nm. According to your suggestion, we have already revised the mistake.

Our modification to the manuscript: The following sentences were modified on pages 7 and 33 of the revised manuscript.

- Page 7

“Upon internalization into tumour cells, the intracellular furin cleaves the RVRRL fragment of CuS-I@P1 NPs, resulting in the exposure of Tyr and release of fluorophore FAM-loaded peptide fragment.”

- Page 33

“b, The time-dependent fluorescence images of MDA-MB-468 tumour bearing mice after intratumour administration of PBS, CuS@P1, CuS-I@P1 and CuS-I@P1-scr.”

Thank you very much for the valuable comments, which have significantly improved the quality of the manuscript.

Reviewers' Comments:

Reviewer #1:

Remarks to the Author:

I am satisfied with the revisions the authors have made. This manuscript can be accepted in its present form.

Reviewer #2:

Remarks to the Author:

The revised manuscript has been improved, and most of the concerns have been addressed effectively. Now, I have no further comments on the manuscript and support it to be published.

Reviewer #3:

Remarks to the Author:

It can be accepted at this stage after the authors addressed my concerns.